# CAN THE TRAINING LOSS BE PREDICTIVE FOR OUT-OF-DISTRIBUTION GENERALIZATION?

## ABSTRACT

Traditional model selection in deep learning relies on carefully tuning several hyper-parameters (HPs) controlling regularization strength on held-out validation data, which can be challenging to obtain in scarce-data scenarios or may not accurately reflect real-world deployment conditions due to distribution shifts. Motivated by such issues, this paper investigates the potential of using solely the training loss to predict the generalization performance of neural networks on out-of-distribution (OOD) test scenarios. Our analysis reveals that preserving consistent prediction variance across training and testing distributions is essential for establishing a correlation between training loss and OOD generalization. We propose architectural adjustments to ensure *variance preservation*, enabling reliable model selection based on training loss alone, even in over-parameterized settings with a sample-to-parameter ratio exceeding four orders of magnitude. We extensively assess the model-selection capabilities of *variance-preserving* architectures on several scarce data, domain-shift, and corruption benchmarks by optimizing HPs such as learning rate, weight decay, batch size, and data augmentation strength.

## 1 INTRODUCTION

The goal of training neural networks is to achieve strong generalization on challenging testing scenarios, which is critical for deploying models in real-world applications where out-of-distribution (OOD) scenarios often arise (Liu et al., 2021). In real-world environments, models are likely to encounter distribution shifts due to a plethora of factors such as corruptions (Hendrycks & Dietterich, 2019) or lack of data (Wad et al., 2022), among others, causing a deviation from the original training distribution.

To ensure generalization, effective regularization techniques are essential, as they are thought to reduce variance and steer the network toward better minima (Foret et al., 2020). These techniques include explicit methods like weight decay (WD) (Zhang et al., 2021a; Andriushchenko et al., 2023), as well as implicit strategies such as using large learning (LR) rates (Lewkowycz et al., 2020; Li et al., 2019) or smaller batch sizes (Keskar et al., 2016; Hoffer et al., 2017). By doing so, they help mitigate overfitting, particularly in scenarios where neural networks are over-parameterized relative to the training set (Advani et al., 2020; Bornschein et al., 2020; Nakkiran et al., 2021; Brigato et al., 2021; 2022).

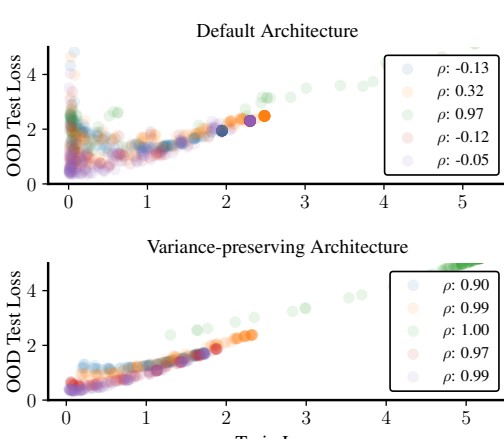

Figure 1: **Predicting OOD generalization.** The training loss of a ResNet-50 is not predictive for OOD generalization when selecting LR and WD (top). Our architecture (bottom) preserves the prediction variance as the test distribution shifts and correlates the training with OOD-test losses despite being severely over-parametrized on CUB and subsampled versions of ISIC 2018, CLaMM, EuroSAT RGB, and EuroSAT.

In real-world applications, practitioners search for a large set of hyper-parameters (HPs) controlling the regularization strength, a crucial step that often determines the model's generalization performance (Krizhevsky et al., 2012; He et al., 2019; Yu & Zhu, 2020). Traditional HP search is performed using validation sets, often by splitting the original training set or by collecting held-out data. Challenges for such model-selection paradigm may arise in use cases where data is: 1) expensive, such as in medical imaging (Varoquaux & Cheplygina, 2022), 2) logistically unfeasible to collect as in federated learning (McMahan et al., 2017), 3) scarce hence inherently unreliable for unbiased evaluations (Lorraine et al., 2020; Brigato & Mougiakakou, 2023), or 4) prone to distribution shifts which often happens in real-world deployment. Concerning the latter point, previous work has investigated ways to measure distances between in-distribution (ID) and OOD distributions (Ben-David et al., 2006) or has shown that linear correlation among ID and OOD test performance may hold (Miller et al., 2021), but not on all cases (Teney et al., 2024). Other approaches may need access to unlabeled OOD data to predict OOD generalization (Tu et al., 2024).

In this work, motivated by the previously mentioned challenges regarding the collection of proper validation data to guide reliable model selection, we raise an unexplored research question (RQ):

**RQ:** *When performing model selection, can we solely rely on the average training loss computed over the ID training set to predict the performance of models on OOD testing scenarios?*

Intuitively, considering only the training loss for model selection seems prohibitive since certain HPs may easily cause over-parametrized models to overfit the ID dataset and consequently obtain poor OOD generalization. To support this claim, we perform a grid search over LR and WD spanning five orders of magnitude ($5 \cdot [10^{-5}, 10^{-1}]$) for a default ResNet-50 (He et al., 2016a) on small datasets with a maximum of 50 samples per class. As expected (see Figure 1, top), we are unable to perform reliable model selection since the training and testing losses are mostly uncorrelated due to multiple configurations scoring a low training loss but a high generalization error.

To address our RQ, we first explore the conditions required for establishing linear relationships between training and test losses as a function of HP choices. From this analysis, we find that the variance of network predictions should remain consistent both within and across train-test distributions. Consequently, we examine how factors such as individual layers, depth, and width scaling influence the ability to preserve prediction variance. Based on these insights, we adapt existing architectures (MLP, ResNet) and configure them to be *variance-preserving* (VP). In other words, we adjust all architectural choices that may enable unbounded variance escalation under distribution shifts. Specifically, we 1) ensure scale-invariance of the function, 2) control variance growth from depth scaling with scaled residuals, and 3) limit variance amplification from width scaling using whitening layers. As visible in Figure 1 (bottom), the effects of our adaptations enable the training loss of the over-parameterized ResNet-50 to serve as a reliable predictor for OOD generalization. **Contribution.** In summary, the contributions of our paper are threefold: 1) **New RQ:** We introduce and explore the paradigm of using the training loss as a reliable predictor of OOD performance for model selection, motivated by the challenges of collecting validation data, especially in scenarios with scarce data or distribution shifts. 2) **Methodology and Architecture Design:** We study the conditions needed to establish linear relationships between training and test losses (Section 2.1) and consequently develop a methodology that controls prediction-variance across distributions by adapting existing architectures to be VP (Section 2.2). **Comprehensive Empirical Analysis:** We analyze the model-selection capabilities of the introduced architectures through an extensive experimental setup (Section 3), including the optimization of several HPs (LR, WD, batch size, and data augmentation strength (Cubuk et al., 2020; Yun et al., 2019; Zhang et al., 2017)) over popular OOD benchmarks covering small-data scenarios (Brigato et al., 2022), domain shifts (Oehri et al., 2024), and corruptions (Hendrycks & Dietterich, 2019; Oehri et al., 2024) benchmarks.

## 2 CAN THE TRAINING LOSS BE PREDICTIVE FOR OOD GENERALIZATION?

### 2.1 ANALYZING CONDITIONS FOR CORRELATING TRAINING AND TEST LOSSES

**Setup**  Let us define a joint $p_{\text{data}}(\text{x}, \text{y})$ and marginal $p_{\text{hp}}(\text{h})$ probability distribution from which we respectively sample training couples $\mathbb{D}_{tr} = \{\boldsymbol{x}, y\}_{i=1}^{n}$, and HP configurations $\mathbb{H} = \{\boldsymbol{h}\}_{i=1}^{h}$. For the sake of simplicity, without loss of generality, in the derivation below, we will focus on tasks where targets are scalars ($y$) rather than vectors ($\boldsymbol{y}$). Let us also define the loss function $\mathcal{L}$,

which measures the discrepancy among the ground truth targets $y$ and the predicted targets $\hat{y}$, with $\hat{y}$ representing the prediction of our learner $f$. Since $f$ is parameterized by $\boldsymbol{w}$ and its learning process is influenced by the sampled hyperparameter (HP) configuration $\boldsymbol{h}$, we define $\hat{y} = f(\boldsymbol{x}, \boldsymbol{w}(\boldsymbol{h}))$. To simplify the relationship between the learned parameters and the HP configuration—reflected in the learning process—we assume that for a fixed architecture-HP-configuration pair, the learning process always converges to a fixed parameter set $\boldsymbol{w}$. While this is clearly a simplifying assumption, it is empirically supported by our results (Section 3.2) and is reasonable under the condition of a fixed optimizer (Section 2.2). In practice, the predictions of a neural network do not vary significantly across repeated runs with the same HP configuration. Therefore, we revisit $\hat{y} = f(\boldsymbol{x}, \boldsymbol{w}(\boldsymbol{h})) \approx f(\boldsymbol{x}, \boldsymbol{h})$ and drop for the sake of our analysis, which focuses on the architecture $f$, the explicit dependence on $\boldsymbol{w}$. The cost over the training distribution given a specific HP configuration $\boldsymbol{h}$ is defined as $J(\boldsymbol{h}) = \mathbb{E}_{\mathrm{x,y}}[\mathcal{L}(\boldsymbol{x}, y, \boldsymbol{h})]$. In practice, we compute the average loss over the training set $\mathbb{D}_{train}$, which means that $J(\boldsymbol{h}) = \frac{1}{n}\sum_i^n \mathcal{L}(y_i, \hat{y}_i)$. We assume to sample the testing set $\mathbb{D}_{te} = \{\boldsymbol{x}, y\}_{i=1}^n$ from another distribution $p'_{\mathrm{data}}(\mathrm{x,y})$ whose marginal distribution $p'(\mathrm{x})$ differs from the original $p(\mathrm{x})$ due to a general covariate shift $p(\mathrm{x}) \to p'(\mathrm{x})$. Our goal is to design the learner $f$ such that the ranking of the cost functions $J$ on the training set $\mathbb{D}_{tr}$ over the sampled HP space $\mathbb{H}$ is consistent with the losses $J'$ over the unknown testing set $\mathbb{D}_{te}$.

**Correlation analysis** To measure the alignment among the two sets of losses and simplify the analysis, we employ the Pearson correlation coefficient $\rho$. In practice, we need $\rho$ to be i) strictly positive since a negative correlation would imply that training and testing losses are negatively correlated and ii) close to one, i.e., $\rho \approx 1$. Let us define the Pearson correlation among training and test losses as $\rho_{J \to J'}$ and show it more formally as:

$$\rho_{J \to J'} = \frac{\mathrm{Cov}(J(\boldsymbol{h}), J'(\boldsymbol{h}))}{\sqrt{\mathrm{Var}(J(\boldsymbol{h})) \cdot \mathrm{Var}(J'(\boldsymbol{h}))}} \tag{1}$$

with $J(\boldsymbol{h}) = \frac{1}{n}\sum_i^n \mathcal{L}(y_i, f(\boldsymbol{x}_i, \boldsymbol{h}))$ and $J'(\boldsymbol{h}) = \frac{1}{n'}\sum_j^{n'} \mathcal{L}(y_j, f(\boldsymbol{x}'_j, \boldsymbol{h}))$. To simplify Equation (1), we break the variance and covariance as a function of the expectation and perform a first-order Taylor expansion of the loss $\mathcal{L}$ around $\boldsymbol{\mu} = \mathbb{E}_{\boldsymbol{h}}(f(\boldsymbol{x}, \boldsymbol{h}))$. Full details of the derivation are provided in Appendix A. The expression for the variance $\mathrm{Var}[J(\boldsymbol{h})]$ and covariance $\mathrm{Cov}[J(\boldsymbol{h}), J'(\boldsymbol{h})]$ of the losses respectively correspond to:

$$\mathrm{Var}(J(\boldsymbol{h})) \approx \frac{1}{n^2}\sum_{i=1}^n\sum_{j=1}^n \mathrm{Cov}_h(\hat{y}_i, \hat{y}_j)\nabla_f\mathcal{L}(y_i, \boldsymbol{\mu}_i)\nabla_f\mathcal{L}(y_j, \boldsymbol{\mu}_j) \tag{2}$$

$$\mathrm{Cov}(J(\boldsymbol{h}), J'(\boldsymbol{h})) = \frac{1}{nn'}\sum_{i=1}^n\sum_{j=1}^{n'} \mathrm{Cov}(\hat{y}_i, \hat{y}'_j)\nabla_f\mathcal{L}(y_i, \boldsymbol{\mu}_i)\nabla_f\mathcal{L}(y_j, \boldsymbol{\mu}'_j) \tag{3}$$

The gradient term $\nabla_f\mathcal{L}$, which appears in both Equations (2) and (3), measures the sensitivity of the loss function for prediction changes against ground truth targets. A large gradient implies that the average loss is very sensitive to small changes in the predictions, amplifying the effect of HP-induced variability on the total variance. Note that since $\mathcal{L}$ is fixed and common to both training and testing evaluation, we can focus our attention on the Cov terms.

$\mathrm{Cov}(\hat{y}_i, \hat{y}'_j)$ measures the variance of the predictions for data points sampled from the distribution $p(\mathrm{x})$ and $p'(\mathrm{x})$ as $\boldsymbol{h}$ change. While $\mathrm{Cov}(\hat{y}_i, \hat{y}_j)$ and $\mathrm{Cov}(\hat{y}'_i, \hat{y}'_j)$ respectively quantify the stability of the model predictions for samples coming from the same distribution, either $p(\mathrm{x})$ or $p'(\mathrm{x})$. To have a high positive correlation among the losses, we need $\rho_{J \to J'} \approx 1$. As specified above, we only consider the covariance terms and the data points $i$ and $j$ to derive:

$$\frac{\mathrm{Cov}(\hat{y}_i, \hat{y}'_j)}{\sqrt{\mathrm{Cov}(\hat{y}_i, \hat{y}_j) \cdot \mathrm{Cov}(\hat{y}'_i, \hat{y}'_j)}} \approx 1 \tag{4}$$

Equation (4) is satisfied if $\text{Cov}(\hat{y}_i, \hat{y}'_j)$ is positive and approximately equal to the denominator. The first condition happens if the model predictions across the different distributions $p(\text{x})$ and $p'(\text{x})$ behave similarly. It implies a *positive* linear relationship among $J(\boldsymbol{h})$ and $J'(\boldsymbol{h})$. The second condition, which determines a *strong positive* linear relationship, happens if the variances of the predictions within each distribution, i.e., $\text{Cov}(\hat{y}_i, \hat{y}_j)$ and $\text{Cov}(\hat{y}'_i, \hat{y}'_j)$ are approximately the same. If the model's predictions are highly stable and aligned within one distribution but more variable in the other, the cross-distribution covariance will not match the product of the within-distribution covariances, thus breaking the approximation.

## 2.2 DESIGNING VARIANCE-PRESERVING ARCHITECTURES

In line with the results of Section 2.1, we aim to develop a *variance-preserving* model whose predictions $f(\boldsymbol{x}, \boldsymbol{h})$, with $\boldsymbol{h} \sim \mathbb{H}$, maintain stability across distribution shifts $p(\text{x}) \rightarrow p'(\text{x})$ and behave similarly within each distribution. Thus, the core objective is to regulate the variance of the network's predictions $\text{Var}(f(\boldsymbol{x}, \boldsymbol{h}))$ during training and adapt neural architectures to minimize the sensitivity to HP variations which can significantly impact the prediction variance.

We focus on several architectural factors that contribute to variance instability, which can result in significant discrepancies between average training and test losses. Note that past literature has extensively studied the phenomenon of distribution shift happening while training, which was defined as *covariate shift* by the seminal paper of Ioffe & Szegedy. In our analysis, we dissect variance shifts along individual layers (Arpit et al., 2016; Li & Arora, 2019; Li et al., 2022), network depth (Glorot & Bengio, 2010; He et al., 2016b; Brock et al., 2021) and network width (Glorot & Bengio, 2010; He et al., 2015; Arpit et al., 2016). Despite this methodology for variance analysis being applicable to different models, we use a 4-layer MLP as a case study to simplify comprehension and provide clearer insights. We progressively modify its architecture to observe the key variance propagation dynamics, leading to the design of a *variance-preserving* network empirically satisfying Equation (1). Before proceeding to the analysis, we address some general notation to describe network architectures and provide more details regarding the specific setup of the chosen case study.

**Notation and setup** Modern deep networks $f(\cdot)$ are modular architectures usually composed of a stack of blocks belonging to three categories: 1) a *stem* layer $s(\cdot)$ which maps input data $\boldsymbol{x}$ to a latent representation $\boldsymbol{z}$, 2) a *trunk* layer $t$ made of identical blocks $g_i$, and 3) an output *head* block $h$ which maps the final representation to the output space. By using the composition notation $\circ$, we summarize the deep network as $f = h \circ t \circ s$, with $t = \bigcirc_{i=1}^{l} g_i$. To simplify notation, we removed the dependence of $f$ to the parameters $\boldsymbol{w}$ and HP $\boldsymbol{h}$. The parameter vector $\boldsymbol{w}$ is trained via SGD optimization to minimize the cross-entropy loss $\mathcal{L}(\boldsymbol{w})$ over $\mathbb{D}_{train}$. The regularized training loss $\mathcal{L}(\boldsymbol{w})_\lambda$ adds to the objective the popular weight decay (WD) term which penalizes the growth of the norm $\boldsymbol{w}$: $\mathcal{L}(\boldsymbol{w})_\lambda = \mathcal{L}(\boldsymbol{w}) + \lambda \frac{||\boldsymbol{w}||^2}{2}$. At each iteration, the parameters follow the well-known update rule $\boldsymbol{w}_{t+1} = \boldsymbol{w}_t - \alpha_t \nabla \mathcal{L}(\boldsymbol{w})_\lambda = \boldsymbol{w}_t - \alpha_t \nabla \mathcal{L}(\boldsymbol{w}) - \alpha_t \lambda \cdot \boldsymbol{w}_t$, where $\alpha_t$ is the learning rate (LR) adjusted at each iteration according to a cosine schedule. Note that we refer to $\alpha$ as the initial LR.

**Case study** We simulate an OOD scenario and keep a substantial difference among $\mathbb{D}_{tr}$ and $\mathbb{D}_{te}$, by randomly sampling 1% of the training dataset of CIFAR-10, maintaining balance across classes, and testing the network on the full test set. The default MLP design includes an identity mapping as stem layer $s = \text{Id}(\cdot)$, a repeated post-activation block $g_i = \text{ReLU}(\text{BN}(\text{Lnr}(\cdot)))$, and a final head $h = \text{Lnr}(\cdot)$. The dimensions of hidden layers are initially fixed to 256, more formally $\boldsymbol{z} \in \mathbb{R}^d$ with $d = 256$. We keep a small batch $b$ of 10 samples, given the tiny size of the training set. As HPs, we sample 10 equally-spaced learning rates $\alpha$ and weight decays $\lambda$ in log-space to perform a full squared-grid search of 100 trials. Each trial is represented by a dot in the scatter plots of this section.

### 2.2.1 CONTROLLING VARIANCE GROWTH OF SINGLE LAYERS WITH SCALE-INVARIANCE

Modern networks using normalization layers such as BN are almost completely scale-invariant (SI), meaning that given a scalar $c \in \mathbb{R}$, $f(c\boldsymbol{w}) \approx f(\boldsymbol{w})$. When trained with SGD and WD, this property gives rise to optimization dynamics, still under investigation (Wan et al., 2021; Kodryan et al., 2022; Andriushchenko et al., 2023), in which the WD does not reduce the complexity of the model but rather increases the *effective learning rate* by reducing the weight norm (Van Laarhoven, 2017; Zhang et al., 2019a), and hence can indirectly exert a regularizing effect by means of larger gradient

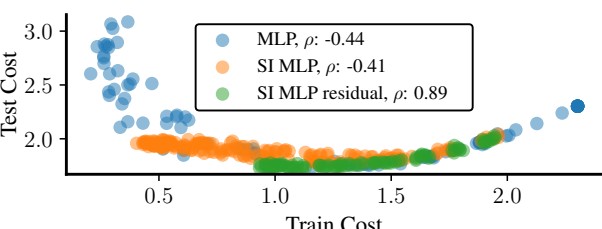

Figure 2: **Scale invariance (SI) and residual connections increase correlation.** SI prevents single layers from exploding in variance, and the residual structure boosts correlation, thanks to linear scaling of variance along depth.

noise (Neelakantan et al., 2015; Keskar et al., 2016; Li et al., 2019; Heo et al., 2020). The gradient updates among scale-variant and scale-invariant layers differ when a critical HP such as WD ($\lambda$) is tuned. For instance, a too-high $\alpha$ coupled with small $\lambda$ could blow the variance of a scale-variant layer due to large gradient updates (Li & Arora, 2019). Thus, we follow previous work to make our MLP SI (Li & Arora, 2019; Kodryan et al., 2022). In practice, we fix $\gamma = 1$ and $\beta = 0$ to prevent BN layers from scaling the variance across network blocks differently and randomly freeze the output layer, which has been shown not to prevent generalization (Hoffer et al., 2018). In Figure 2, we see that SI prevents the unbounded growth of the test losses when the training cost is very low, i.e., when strong overfitting happens. However, the correlation coefficient is still negative, meaning that predictions behave differently among $p(\mathrm{x})$ and $p'(\mathrm{x})$.

### 2.2.2 PREVENTING VARIANCE INCREASE DUE TO DEPTH SCALING WITH SCALED RESIDUALS

A further reason for this misalignment is represented by the bad propagation of the signal in the forward-backward passes induced by the MLP architecture. Indeed, the variance across the layers, as widely studied in previous initialization literature Glorot & Bengio, 2010; He et al., 2015, scales as the product of depth. The sequence of matrix-vector products is hence highly unstable when large gradient noise comes from our strong data-distribution shift and broad HP space, causing vanishing or exploding gradients.

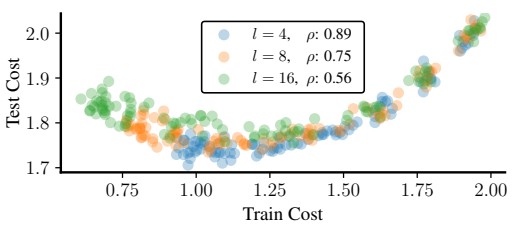

Figure 3: **Depth scaling reduces correlation.** Given the linear growth in variance, default additive residual connections reduce correlation as the network depth $l$ increases.

To better preserve the signal variance (Taki, 2017), we indeed employ the popular residual connections popularized by the ResNet architecture and hence *ResNet-ctify* the MLP. Furthermore, we move from pre- to post-activation (He et al., 2016b), which further simplifies the propagation of the signal thanks to the identity skip connection. As visible in Figure 2 (green), the residual design significantly benefits alignment among training and test losses by raising $\rho$ from -0.44 to 0.89. Thus, we modify the MLP architecture components $s = \mathrm{Lnr}(\cdot)$, $g_i = \mathrm{Lnr}(\mathrm{ReLU}(\mathrm{BN}(\cdot)))$, and $h = \mathrm{Lnr}(\mathrm{ReLU}(\mathrm{BN}(\cdot)))$. The hidden activation $\boldsymbol{z}_i$ is now computed with the additive residual connection $\boldsymbol{z}_{i+1} = g(\boldsymbol{z}_i) + \boldsymbol{z}_i$.

However, additive residual connections cause the variance to increase linearly with depth, as each addition contributes to the overall variance (Zhang et al., 2019b; Brock et al., 2021; Hoedt et al., 2022). Formally, in networks with additive residual connections, the variance at the $i_{th}$ block of the *trunk* becomes $\mathrm{Var}(\boldsymbol{z}_{i+1}) = \mathrm{Var}(g_i(\boldsymbol{z}_i)) + \mathrm{Var}(\boldsymbol{z}_i)$. To empirically visualize the variance growth and quantify its impact on the alignment among train and test losses, we compare the 4-layer residual MLP against the 8- and 16-layer versions. The correlation coefficient drops from 0.89 with 4 layers to 0.56 with 16 layers (Figure 3).

To prevent variance explosion, we apply techniques such as scaling the residual branch by a factor $\delta$, with $\delta = l^{-1}$, to enable stable variance propagation in a network with $l$ skip connections (Arpit et al., 2019). We hence dampen the variance contribution from the residual path as $\boldsymbol{z}_{i+1} = \delta g_i(\boldsymbol{z}_i) + \boldsymbol{z}_i$. We employ the Signal Propagation Plots (SPPs) introduced by (Brock et al., 2021) and scatter the average variance per activation $\mathrm{Var}(\boldsymbol{z})$ when $\boldsymbol{x} \sim \mathcal{N}(0, 1)$. In the SPP of Figure 4 (right), we appreciate the propagation of variance under control for the scaled residual MLP with 16 layers

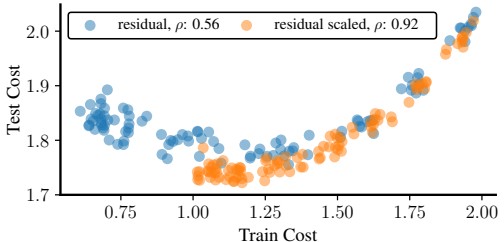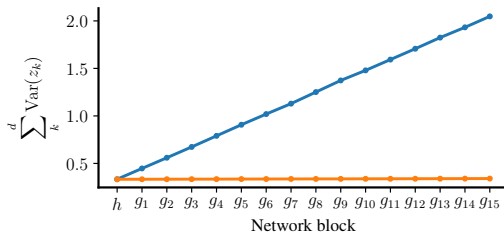

Figure 4: **Scaled residuals preserve variance increase due to depth scaling.** The variance grows linearly in the case of default residual connections but remains constant if we scale the residual by a factor $l^{-1}$ (right). We see the direct effect of controlling variance in increased correlation (left).

against the default linear growth. This design translates into alignment among train and test losses ($\rho = 0.92$), visible in Figure 4 (left).

## 2.3 LIMITING VARIANCE ESCALATION DUE TO WIDTH SCALING WITH GROUP WHITENING

Next, we study the impact of width on the alignment between train and test losses. Considering a pre-activation element $z_{i,k}$ of any linear layer $\mathrm{Lnr}(\cdot)$, we compute the variance of post-activation elements $z_{i+1,k}$ after the multiplication with the parameters $W_{k,j}$. More precisely, $\mathrm{Var}(z_{i+1,k}) = \mathrm{Var}(\sum_j^d W_{k,j} z_j)$. Let us define $a_j = W_{k,j} z_j$ and expand the variance of sums to obtain $\mathrm{Var}(z_{i+1,k}) = \sum_j^d \mathrm{Var}(a_j) + \sum_{j \neq k}^d \mathrm{Cov}(a_j, a_k)$. If we assume constant variances $\sigma_a^2$ and covariances $\kappa_a^2$, we get $\mathrm{Var}(z_{i+1,k}) = d\,\sigma_a^2 + d(d-1)\,\kappa_a^2$. Thus, the pre-activation covariances quadratically scale the post-activation variance as a function of width. It is known that correlation

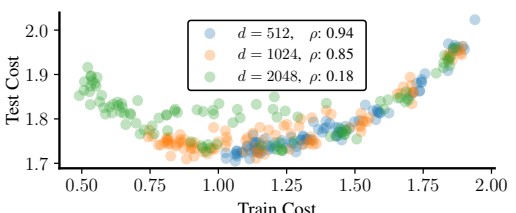

Figure 5: **Width scaling reduces correlation.** Cross-correlations in pre-activations scale post-activation variance quadratically, thus correlation decreases sharply as width $d$ increases.

among hidden activations, generally assumed to be zero for initialization schemes (Glorot & Bengio, 2010; He et al., 2015), may drastically grow during training since BN layers only standardize activations (Ioffe & Szegedy, 2015; Arpit et al., 2016). As the network grows in size, activations may increase their correlation due to the redundancy in learned representations (Morcos et al., 2018; Kornblith et al., 2019). We empirically visualize the increase of variance by increasing the size of width from $d = 512$ to $d = 2048$ in powers of 2. As visible in Figure 5, as we increase the width $d$, the alignment drastically diminishes, with $\rho$ dropping from 0.94 to 0.18.

A way to counteract this effect, which yet presents several challenges, is to whiten the activations. A straightforward way is to apply a penalty term to the activation covariances, as investigated in previous work (Cogswell et al., 2015; Hua et al., 2021). However, this approach would add extra HPs that we are willing to avoid, given our RQ. We hence refer to another line of work which have proposed batch whitening algorithms in deep networks (Huang et al., 2018; 2019; 2020; Siarohin et al., 2018). Two key challenges of such methods regard i) the increased computational complexity, which scales as $\mathcal{O}(d^2 \max(b, d))$ in the full-batch case (Huang et al., 2018) and ii) instability due to matrix inversion and small batches (Huang et al., 2019; 2020).

As a solution, we substitute the BN layer of the *head* block with a Group Whitening (GW) layer (Huang et al., 2021) and leave the rest of the normalization layers with BN. Therefore the *head* of our residual MLP is modified to $h = \mathrm{Lnr}(\mathrm{ReLU}(\mathrm{GW}(\cdot)))$. This design has three main advantages: 1) it leaves the computational cost practically unaffected, 2) it is independent of $b$, which may be problematic in small-batch settings, and 3) it whitens the activation of the full network via the backward pass despite using a single layer thanks the residual connections. To show the latter point, let $z_0 = s(x)$ be the output of the stem layer and let $z_l = z_0 + \sum_{i=1}^l g_i(z_{i-1})$ following our network structure. If we compute the derivative of $\mathrm{GW}(z_l)$ with respect to $z_i$ we get:

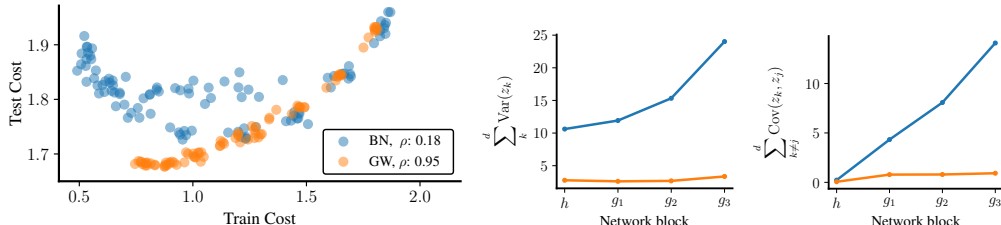

Figure 6: **Group whitening preserves variance growth due to width scaling.** The variance grows due to high average covariances in the activation vectors but remains constant if we substitute the batch normalization (BN) layer with group whitening (GW) in the *head* of the network (center and right). The correlation increases if we control variance growth in this manner (left).

$$\frac{\partial \mathrm{GW}(\boldsymbol{z}_l)}{\partial \boldsymbol{z}_i} = \frac{\partial \mathrm{GW}(\boldsymbol{z}_l)}{\partial \boldsymbol{z}_l} \cdot \left(1 + \frac{\partial}{\partial \boldsymbol{z}_i} \sum_{k=i}^{l-1} g_k(\boldsymbol{z}_k)\right) \tag{5}$$

We indeed note that the derivative of the whitening layer $\frac{\partial \mathrm{GW}(\boldsymbol{z}_l)}{\partial \boldsymbol{z}_l}$, containing the group-wise inverted covariances (Huang et al., 2021), scales the gradients of all network blocks $\frac{\partial g_k(\boldsymbol{z}_k)}{\partial \boldsymbol{z}_i}$. We directly see its effect on the large 2048-wide MLP trained with high LR and low WD ($\alpha = 0.5, \lambda = 5 \cdot 10^{-5}$). In the SPPs of Figure 6 (center and right), we feed the training set to the trained networks with or without the GW layer and plot the average variances and upper-diagonal covariances. By keeping the average covariance of activations close to zero (Figure 6, right), GW prevents the growth of variance due to width scaling (Figure 6 center). We hence experience an almost perfect alignment ($\rho = 0.95$), as visible in Figure 6 (left), despite the network being trained on as few as 500 examples and containing approximately 20M parameters.

## 3 EXPERIMENTS

In this section, we validate and evaluate the ability to align train and test losses and the generalization capabilities of *variance-preserving* architectures in more challenging test cases. Thus, we describe the experimental setup in Section 3.1. We discuss the empirical results concerning alignment in Section 3.2 and the quality of model selection in terms of absolute performance in Section 3.3.

### 3.1 EXPERIMENTAL SETUP

**OOD benchmarks**   To cover a broad spectrum of real-world use cases, we focus on three types of OOD scenarios: 1) small datasets, 2) corruptions, and 3) domain shifts.

*Small datasets* challenge the IID assumption of standard machine learning and hence are considered an OOD testbed (Wad et al., 2022). We select the benchmark introduced in (Brigato et al., 2022) containing five different datasets spanning various domains and data types. In particular, the benchmark sub-sampled ciFAIR-10 (c10) (Barz & Denzler, 2020), EuroSAT (ES) (Helber et al., 2019), CLaMM (CLM) (Stutzmann, 2016), all with 50 samples per class, and ISIC 2018 (ISIC), with 80 samples per class (Codella et al., 2019). The popular CUB (Wah et al., 2011), with 30 images per category, is the last dataset of the benchmark. The spanned image domains of this benchmark hence include RGB natural images (c10, CUB), multi-spectral/RGB satellite data (ES, ESR), RGB skin medical imagery (ISIC), and grayscale hand-written documents (CLM).

*Corruptions* of various types have been introduced by (Hendrycks & Dietterich, 2019) to measure the robustness of deep networks to common corruptions. We employ the popular CIFAR10-C (C10C) and CIFAR100-C (C100C), where the original test datasets are subjected to 15 different corruptions, each at five severity levels. For a more challenging scenario, we also employ the recently introduced corrupt versions of TinyImagenet-C (TINC) and EuroSATRGB-C (ESRC) from (Oehri et al., 2024).

| HPS | 1 | 1 | 1 | 1 | 1 | 1 | 1 | 1 | 1 | 3 | 3 | 3 | 3 | HPS |
|---|---|---|---|---|---|---|---|---|---|---|---|---|---|---|
| | | | **Small datasets** | | | | **Corruptions** | | | **Domain shifts** | | | | |
| | c10 | ISIC | CLM | CUB | ES | ESR | ESRC | C10C | C100C | TINC | TINV2 | TINR | TINA | |
| **Arch.** | | | | | | Pearson Correlation ($\rho$) | | | | | | | | **Avg.** |
| Def. | 0.34 | -0.12 | 0.22 | 0.97 | -0.01 | -0.12 | -0.21 | 0.85 | 0.83 | 0.70 | 0.90 | -0.21 | 0.81 | 0.38 |
| VP | 0.94 | 0.90 | 0.99 | 1.00 | 0.99 | 0.97 | 0.75 | 1.00 | 1.00 | 0.87 | 0.92 | 0.72 | 0.85 | 0.92 |
| **Arch.** | | | | | | Spearman Rank Correlation ($\rho_s$) | | | | | | | | **Avg.** |
| Def. | 0.19 | -0.03 | 0.51 | 0.82 | 0.54 | 0.45 | -0.37 | 0.87 | 0.83 | 0.81 | 0.96 | -0.32 | 0.89 | 0.47 |
| VP | 0.82 | 0.96 | 0.99 | 1.00 | 0.96 | 0.95 | 0.69 | 0.99 | 1.00 | 0.96 | 0.98 | 0.77 | 0.97 | 0.93 |
| **Arch.** | | | | | | Weighted Kendall Rank Coefficient ($\tau_w$) | | | | | | | | **Avg.** |
| Def. | 0.06 | -0.09 | 0.29 | 0.40 | 0.31 | 0.23 | -0.34 | 0.86 | 0.82 | 0.72 | 0.94 | -0.01 | 0.76 | 0.38 |
| VP | 0.90 | 0.93 | 0.96 | 0.96 | 0.93 | 0.91 | 0.82 | 0.98 | 0.97 | 0.93 | 0.94 | 0.37 | 0.92 | 0.89 |

Table 1: **Alignment between train and OOD test losses.** The VP architecture exhibits a strong correlation between train and test loss and outperforms the default architecture across metrics.

*Domain shifts* of several types represent a challenging OOD scenario for deep networks. We gather the Tiny ImageNet test sets featuring the popular distribution-shift benchmarks of ImageNet, recently introduced by (Oehri et al., 2024). More specifically, Tiny ImageNetV2 keeps all images of joint classes of Tiny ImageNet and ImageNetV2 (Recht et al., 2019). Similarly, Tiny ImageNet-R benchmarks the robustness of models when confronted with domain shifts, such as changes in the type of images (e.g., paintings, toys, or graffiti). Finally, Tiny ImageNet-A contains all images from the original Tiny ImageNet validation set misclassified by a ResNet-18.

**Architectures** As the main architecture, we focus on ResNet-50 (RN50) (He et al., 2015), given its large popularity and great adaptability for tasks of small-to-medium size. For images or resolutions smaller than 64, we employ either the Wide ResNet (WRN) (Zagoruyko & Komodakis, 2016) or the previously introduced MLP. Given the simplicity of the VP adaptation discussed in Section 2.2, it is straightforward to apply such adjustments to the RN50 or low-resolution WRN. The only key difference regards the addition of BN layers on the skip connections where down-sampling happens to keep the SI property of the network (Li & Arora, 2019; Kodryan et al., 2022).

**Training setup** We mostly train 100 models per run up to 200 to simulate an extensive HP search. We employ SGD for *variance-preserving* and SGDM for default architectures. The latter is set with momentum $\mu$ equal to 0.9 as standard practice. We employ grid and random search strategies, as well as early-stopping schedulers such as Asynchronous Successive Halving Algorithm (ASHA) (Li et al., 2020a), to both decrease computational demand and increase real-world conditions. We test multiple HP setups (HPS) of varying difficulties and breadth starting from $\alpha$ and $\lambda$ (HPS1), being the two most popular HPs searched by practitioners. We then add to HPS1 the batch size $b$ (HPS2) and, finally, the data augmentation strength (HPS3). In particular, we search for the HPs $N$ and $M$ from RandAugment (Cubuk et al., 2020), the Beta distribution parameters $\lambda_{\mathrm{mu}}$ of MixUp and $\lambda_{\mathrm{cm}}$ of CutMix (Yun et al., 2019), and probability $p_{\mathrm{mu}}$ of applying MixUp. For additional details regarding the training details and HPSs, refer to Appendix B.

**Metrics** To validate the functional relationship between train and test losses, we add to the previously mentioned $\rho$, which measures linear correlation, the Spearman's rank correlation coefficient $\rho_s \in [-1, 1]$ to measure monotonicity. In addition, we also compute the weighted variant of Kendall's rank correlation $\tau_w \in [-1, 1]$, which is often employed to measure when selecting the best-ranked item of interest (You et al., 2021; Tu et al., 2024). To validate the recognition performance, we employ the test loss and the test accuracy computed on the OOD test set. For imbalanced test sets, we compute the balanced accuracy, i.e., the average of the per-class accuracies.

## 3.2 ASSESSING THE FUNCTIONAL RELATIONSHIP AMONG TRAIN AND TEST LOSSES

Here, we validate the capability of the VP networks to withstand distribution shifts compared to the default architectures. In Table 1, we observe that the VP architectures maintain a high correlation

| HPS | | | 2 | 2 | 2 | 2 | 2 | 1 | 1 | 1 | 1 | 3 | 3 | 3 | 3 |
|---|---|---|---|---|---|---|---|---|---|---|---|---|---|---|---|
| | | | | Small datasets | | | | | Corruptions | | | | Domain shifts | | |
| | | | c10 | ISIC | CLM | CUB | ES | ESR | ESRC | C10C | C100C | TINC | TINV2 | TINR | TINA |
| Arch. | Val | | Recognition Performance (%) | | | | | | | | | | | | |
| Def. | ✓ | 55.2* | 64.5* | 70.2* | 70.8* | 90.6* | 82.5 | 59.8 | 62.1 | 35.8 | 29.2 | 38.1 | 13.7 | 21.5 |
| VP | ✗ | 57.1 | 70.8 | 46.2 | 61.6 | 91.3 | 81.5 | 55.5 | 58.8 | 25.8 | 17.8 | 30.1 | 8.7 | 16.1 |
| VP+ | ✗ | 60.3 | 67.6 | 70.5 | 64.5 | 91.6 | 82.0 | 56.4 | 60.8 | 28.2 | 18.2 | 30.3 | 9.01 | 15.6 |

Table 2: **Generalization on OOD benchmarks.** VP+ architecture performs comparably to their default counterparts on tasks with $\leq 15$ classes, achieving similar average scores (69.9% vs 69.3%) without needing validation. On tasks with 100+ classes, it underperforms due to constraints prioritizing decorrelated representations and may require lower regularization and higher learning rates. *The values are reported from the benchmark in (Brigato et al., 2022).

between train and OOD test loss despite the kind of distribution shift. On average, over the 13 cases we tested, VP architectures record a strong positive correlation for all three indices with $\rho = 0.92$, $\rho_s = 0.93$, and $\tau_w = 0.89$. On the other hand, base architectures show difficulties in maintaining a solid alignment, recording a linear correlation $\rho$ of 0.38, monotonicity $\rho_s$ of 0.47, and a weighted Kendall coefficient $\tau_w$ of 0.38. Note that this evaluation concerns heterogeneous setups that include the HPs spaces ranging from the grid search of LR and WD (Small Datasets, ESRC, C10C, C10C) to the more complex random search with ASHA scheduler with the addition of batch size and data augmentation parameters in the case of TINC, and domain shift datasets. In Appendix C.1, we analyze the characteristics of our tested distribution shifts following (Ye et al., 2022). In Appendix C.2, we show that VP architectures better handle distribution shifts of increasing magnitude.

### 3.3 Assessing Generalization of Variance-preserving Architectures

After showing the predictive power for OOD generalization of the introduced architecture, we test its absolute generalization capabilities. More specifically, we compare the VP architecture, solely optimized according to its training loss, against networks trained with the traditional training/validation paradigm. One of the key advantages of VP networks is that they eliminate the need for data splitting and re-training. Leveraging this property, we optionally fine-tune the best configuration identified during the HP search with an additional run on the same training set (VP+). While this second training step might seem redundant in our setup, it is relevant to note that in the traditional train-validation-split paradigm, such a re-training step is always required.

In Table 2, we observe that VP+ architectures perform comparably well (69.9% vs 69.3%) to their default counterparts on tasks having $\leq 15$ classes despite not necessitating the external validation signal from held-out data. In line with observations from previous studies, a further fine-tuning step might be needed because SI architectures necessitate slightly longer training schedules to reach the same performance due to the inherent constraints within the network architecture (Li et al., 2022). On tasks with a more significant number of classes (e.g., $\geq 100$), such as CUB, C100C, and the TIN variants, the VP architecture tends to face more challenges. This is likely due to the architectural constraints designed to preserve decorrelated representations, which make it more difficult to distinguish between many classes with fine-grained differences at the lower layers of the network. This suggests that while the variance-preservation mechanism supports certain advantages, such as the strong predictive performance of OOD generalization from the training loss as seen in Table 1, it may introduce trade-offs when scaling to more complex classification tasks. Additionally, we observed that the optimal HP range, typically effective for default architectures, may shift toward lower regularization and higher learning rates, further highlighting the need for longer training and reduced regularization inherent to the VP architecture.

## 4 Related Work

**Predicting generalization** Most of the work on predicting generalization in deep learning has focused on assessing the generalization gap, i.e., the difference between train and test generalization, via several complexity metrics based on model parameters, the training set, and distributional ro-

bustness, among others (Keskar et al., 2016; Chuang et al., 2021; Smith & Le, 2017; Dziugaite & Roy, 2017; Dinh et al., 2017; Dziugaite et al., 2020; Jiang et al., 2019; Corneanu et al., 2020; Jiang et al., 2018; Neyshabur et al., 2017). Our work differs since 1) it focuses on scenarios where the IID assumption does not hold, and 2) it utilizes the training loss as a predictor of generalization. A vast literature has focused on tackling OOD generalization (Liu et al., 2021; Sagawa et al., 2019; Zhang et al., 2021b; Huang et al., 2022) and ways to benchmark it (Hendrycks & Dietterich, 2019; Koh et al., 2021; Oehri et al., 2024; Vedantam et al., 2021). Standard model selection for OOD methods employs validation data by either splitting samples from all environments or leaving one environment out (Gulrajani & Lopez-Paz, 2020). Several works studied how to maintain consistent variance across training environments to improve generalization, including setting penalties for weighting training or validation risks (Ye et al., 2021; Krueger et al., 2021; Arjovsky et al., 2019), or generating synthetic new environments through generative modeling (Bai et al., 2021). (Sagawa et al., 2019) studied how regularization improves worst-group performance. Unlike all these works, we do not rely on any validation signal but only the training loss to perform model selection. More related to our work, past research has tried to predict the OOD generalization from ID performance, assuming access to a labeled test dataset sampled from the same training distribution (Ben-David et al., 2006; Tachet des Combes et al., 2020; Miller et al., 2021). Others have focused on the prediction problem, assuming they can access the unlabeled OOD test set to compute relevant prediction metrics (Deng & Zheng, 2021; Deng et al., 2022; Peng et al., 2023; Teney et al., 2024). Our work differs from both lines, given that we are predicting the OOD performance solely employing the ID training loss.

**Hyper-parameter tuning.** There is a vast literature tackling the problem of HP tuning for deep networks (Yu & Zhu, 2020), including works on implicit differentiation (Lorraine et al., 2020), data augmentation (Cubuk et al., 2019; Li et al., 2020b), neural-architecture search (Elsken et al., 2019), invariance learning (van der Wilk et al., 2018; Benton et al., 2020; Immer et al., 2022), and general-purpose schedulers (Li et al., 2017; 2020a). Concerning optimization-related HPs, the seminal work of Goyal et al. (Goyal et al., 2017) popularized the linear scaling rule for learning rate and batch size. Recent research proposed parameterization to transfer LRs to larger model sizes (Yang et al., 2021; Everett et al., 2024). Recent work studied HP selection as data scales by exploiting SGD symmetries (Yun et al., 2020; 2022). However, only a few studies explore HP optimization without employing validation sets, mainly focusing on learning invariances. When employing Bayesian inference, methods either fail to scale to relatively simple tasks (e.g., CIFAR-10) (Schwöbel et al., 2022) or larger network sizes (e.g., ResNet-14) (Immer et al., 2022). Benton et al. (Benton et al., 2020) make strong assumptions about knowing what HPs help learning invariances in advance. A recent method improves scalability issues but still introduces complexity by needing data and model partitioning and an additional backward-forward pass (Mlodozeniec et al., 2023). Unlike such methods, we focused on predicting generalization from the training loss without setting limits to HP types, proposing a simple architectural adaptation.

## 5 CONCLUSIONS

This paper introduces the unexplored RQ of utilizing the training loss as an indicator for ranking OOD performance in neural networks, motivated by the difficulty of collecting reliable validation data for real-world scenarios. We derive the importance of maintaining consistent prediction variance across training and testing distributions to establish a correlation with OOD generalization. Through our analysis, we identify the architectural adjustments necessary for achieving variance preservation, thereby enabling model selection over a broad HP space based solely on training loss, even in OOD over-parameterized scenarios. Our extensive empirical validation, conducted across 13 OOD benchmarks, demonstrates that VP architectures enable strong predictability of generalization with comparable classification performance on datasets with a small number of classes. In summary, our contributions lay the groundwork for a new class of architectures that eliminates reliance on validation data and promotes training loss as a robust indicator of OOD performance. Future work will focus on adapting and testing the design on other models (e.g., vision transformer (Dosovitskiy et al., 2020)) and close the remaining performance gap on datasets with more classes.

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

# A COMPUTATION OF VARIANCE AND COVARIANCE AMONG TRAIN AND TEST LOSSES

**Variance computation.** The variance of the average loss $J$ is defined as:

$$\mathrm{Var}(J(\boldsymbol{h})) = \mathbb{E}(J(\boldsymbol{h})^2) - \mathbb{E}(J(\boldsymbol{h}))^2 \tag{6}$$

Computation of $\mathbb{E}(J(\boldsymbol{h}))^2$. We first compute the second term from Equation (6). The expected value of $J(\boldsymbol{h})$ is:

$$\mathbb{E}(J(\boldsymbol{h})) = \frac{1}{n} \sum_{i=1}^{n} \mathbb{E}(\mathcal{L}(y_i, f(\boldsymbol{x}_i, \boldsymbol{h}))) \tag{7}$$

For simplicity of derivation, using a first-order Taylor expansion of the loss function around the mean prediction $\boldsymbol{\mu}_i = \mathbb{E}(f(\boldsymbol{x}_i, \boldsymbol{h}))$, we get:

$$\mathcal{L}(y_i, f(\boldsymbol{x}_i, \boldsymbol{h})) \approx \mathcal{L}(y_i, \boldsymbol{\mu}_i) + \nabla_f \mathcal{L}(y_i, \boldsymbol{\mu}_i) \cdot (f(\boldsymbol{x}_i, \boldsymbol{h}) - \boldsymbol{\mu}_i) \tag{8}$$

Taking the expectation with respect to $\boldsymbol{h}$, we get:

$$\mathbb{E}(\mathcal{L}(y_i, f(\boldsymbol{x}_i, \boldsymbol{h}))) \approx \mathcal{L}(y_i, \boldsymbol{\mu}_i) \tag{9}$$

Thus, we substitute back to get $\mathbb{E}(J(\boldsymbol{h}))$ and $\mathbb{E}(J(\boldsymbol{h}))^2$:

$$\mathbb{E}(J(\boldsymbol{h})) \approx \frac{1}{n} \sum_{i=1}^{n} \mathcal{L}(y_i, \boldsymbol{\mu}_i), \quad \mathbb{E}(J(\boldsymbol{h}))^2 \approx \left( \frac{1}{n} \sum_{i=1}^{n} \mathcal{L}(y_i, \boldsymbol{\mu}_i) \right)^2 \tag{10}$$

Computation of $\mathbb{E}(J(\boldsymbol{h})^2)$. Now, we compute the expectation of $J(\boldsymbol{h})^2$:

$$J(\boldsymbol{h})^2 = \frac{1}{n^2} \sum_{i=1}^{n} \sum_{j=1}^{n} \mathcal{L}(y_i, f(\boldsymbol{x}_i, \boldsymbol{h})) \mathcal{L}(y_j, f(\boldsymbol{x}_j, \boldsymbol{h})) \tag{11}$$

Taking the expectation with respect to $\boldsymbol{h}$:

$$\mathbb{E}(J(\boldsymbol{h})^2) = \frac{1}{n^2} \sum_{i=1}^{n} \sum_{j=1}^{n} \mathbb{E}(\mathcal{L}(y_i, f(\boldsymbol{x}_i, \boldsymbol{h})) \mathcal{L}(y_j, f(\boldsymbol{x}_j, \boldsymbol{h}))) \tag{12}$$

Using the first-order Taylor expansion for both terms:

$$\mathcal{L}(y_i, f(\boldsymbol{x}_i, \boldsymbol{h})) \approx \mathcal{L}(y_i, \boldsymbol{\mu}_i) + \nabla_f \mathcal{L}(y_i, \boldsymbol{\mu}_i) \cdot (f(\boldsymbol{x}_i, \boldsymbol{h}) - \boldsymbol{\mu}_i) \tag{13}$$

We multiply the expanded terms (linear approximation), and take the expectation over $\boldsymbol{h}$. Since the cross terms involving $(f(\boldsymbol{x}_i, \boldsymbol{h}) - \boldsymbol{\mu}_i)$ vanish when taking the expectation, we obtain that $\mathbb{E}(\mathcal{L}(y_i, f(\boldsymbol{x}_i, \boldsymbol{h})) \mathcal{L}(y_j, f(\boldsymbol{x}_j, \boldsymbol{h})))$ is:

$$\approx \mathcal{L}(y_i, \boldsymbol{\mu}_i) \mathcal{L}(y_j, \boldsymbol{\mu}_j) + \mathrm{Cov}(f(\boldsymbol{x}_i, \boldsymbol{h}), f(\boldsymbol{x}_j, \boldsymbol{h})) \nabla_f \mathcal{L}(y_i, \boldsymbol{\mu}_i) \nabla_f \mathcal{L}(y_j, \boldsymbol{\mu}_j) \tag{14}$$

Thus:

$$\mathbb{E}(J(\boldsymbol{h})^2) \approx \frac{1}{n^2} \sum_{i=1}^{n} \sum_{j=1}^{n} \big( \mathcal{L}(y_i, \boldsymbol{\mu}_i) \mathcal{L}(y_j, \boldsymbol{\mu}_j)$$
$$+ \mathrm{Cov}(f(\boldsymbol{x}_i, \boldsymbol{h}), f(\boldsymbol{x}_j, \boldsymbol{h})) \nabla_f \mathcal{L}(y_i, \boldsymbol{\mu}_i) \nabla_f \mathcal{L}(y_j, \boldsymbol{\mu}_j) \tag{15}$$

Final variance expression. Substituting the previous results into Equation (6), we get:

$$\mathrm{Var}(J(\boldsymbol{h})) \approx \frac{1}{n^2} \sum_{i=1}^{n} \sum_{j=1}^{n} \mathrm{Cov}(f(\boldsymbol{x}_i, \boldsymbol{h}), f(\boldsymbol{x}_j, \boldsymbol{h})) \nabla_f \mathcal{L}(y_i, \boldsymbol{\mu}_i) \nabla_f \mathcal{L}(y_j, \boldsymbol{\mu}_j) \tag{16}$$

**Covariance computation.** The covariance between two cost functions $J(\boldsymbol{h})$ and $J'(\boldsymbol{h})$ is given by:

$$\mathrm{Cov}(J(\boldsymbol{h}), J'(\boldsymbol{h})) = \mathbb{E}(J(\boldsymbol{h})J'(\boldsymbol{h})) - \mathbb{E}(J(\boldsymbol{h}))\mathbb{E}(J'(\boldsymbol{h})) \tag{17}$$

Computation of $\mathbb{E}(J(\boldsymbol{h})J'(\boldsymbol{h}))$ Expanding the product:

$$J(\boldsymbol{h})J'(\boldsymbol{h}) = \frac{1}{nn'} \sum_{i=1}^{n} \sum_{j=1}^{n'} \mathcal{L}(y_i, f(\boldsymbol{x}_i, \boldsymbol{h})) \mathcal{L}(y_j', f(\boldsymbol{x}_j', \boldsymbol{h})) \tag{18}$$

Taking the expectation with respect to $\boldsymbol{h}$:

$$\mathbb{E}(J(\boldsymbol{h})J'(\boldsymbol{h})) = \frac{1}{nn'} \sum_{i=1}^{n} \sum_{j=1}^{n'} \mathbb{E}(\mathcal{L}(y_i, f(\boldsymbol{x}_i, \boldsymbol{h})) \mathcal{L}(y_j', f(\boldsymbol{x}_j', \boldsymbol{h}))) \tag{19}$$

By using again the first-order Taylor expansion for both losses and simplifying the cross terms, the expectation $\mathbb{E}(J(\boldsymbol{h})J'(\boldsymbol{h}))$ becomes:

$$\frac{1}{nn'} \sum_{i=1}^{n} \sum_{j=1}^{n'} \big( \mathcal{L}(y_i, \boldsymbol{\mu}_i) \mathcal{L}(y_j', \boldsymbol{\mu}_j') + \mathrm{Cov}(f(\boldsymbol{x}_i, \boldsymbol{h}), f(\boldsymbol{x}_j', \boldsymbol{h})) \nabla_f \mathcal{L}(y_i, \boldsymbol{\mu}_i) \nabla_f \mathcal{L}(y_j', \boldsymbol{\mu}_j') \big) \tag{20}$$

Final covariance expression: Substituting the expected values $\mathbb{E}(J(\boldsymbol{h}))$ previously computed, we derive the final expression for the covariance:

$$\mathrm{Cov}(J(\boldsymbol{h}), J'(\boldsymbol{h})) = \frac{1}{nn'} \sum_{i=1}^{n} \sum_{j=1}^{n'} \mathrm{Cov}(f(\boldsymbol{x}_i, \boldsymbol{h}), f(\boldsymbol{x}_j', \boldsymbol{h})) \nabla_f \mathcal{L}(y_i, \boldsymbol{\mu}_i) \nabla_f \mathcal{L}(y_j', \boldsymbol{\mu}_j') \tag{21}$$

# B TRAINING SETUP

Here, we provide more details regarding the three different hyper-parameter setups (HPS) employed, along with additional training details.

## B.1 IMPLEMENTATION DETAILS AND HYPER-PARAMETER SETUP 1 (HPS1)

**Small datasets** In this setup, we optimize LR and WD. In particular, for all datasets, we sample 10 equally-spaced learning rates $\alpha$ and weight decays $\lambda$ in log-space to perform a full squared-grid search of 100 trials with no early stopping. More precisely $\boldsymbol{h} = [\alpha, \lambda] \sim \mathrm{LogUniform}(5 \cdot [10^{-5}, 10^{-1}], 10)$. In this case, due to the absence of randomness from sampling and preliminary

experiments indicating minimal variations due to weight initializations, we conduct a single run. For the small datasets, we train all networks with batch sizes of 10 samples, given the better generalization performance of small batch sizes in small-sample regimes (Brigato et al., 2021; 2022). The training iterations are drawn from (Brigato et al., 2022), with a minimum of 25,000 for the smaller datasets and a maximum of roughly 120,000 for CUB. We employ standard image pre-processing transformations, including data augmentation utilizing random crops and horizontal flipping with different strength depending on the specific dataset, also following (Brigato et al., 2022). More precisely, all input images were normalized by subtracting the channel-wise mean and dividing by the standard deviation computed on the training splits. For datasets with a small, fixed image resolution, i.e., ciFAIR-10, EuroSAT, and EuroSAT RGB, we perform random shifting by 12.5% of the image size and horizontal flipping in 50% of the cases. For all other datasets, we apply scale augmentation using the `RandomResizedCrop` transform from PyTorch[1]. For these experiments, we employ a RN50 for all datasets with resolution $\geq 64{\times}64$ and a WRN-16-10 for ciFAIR-10.

**CIFAR-10C and 100C**  We again optimize for LR and WD. We followed the same augmentation strategy described in the previous paragraph for ciFAIR-10. We fixed the batch size to 50 samples instead. We also set the number of epochs to 100, hence training the models for 100,000. For these experiments, we employ the MLP described in Section 2.2 with a depth of 4 and a width of 2048 (MLP-4-2048).

### B.2  Implementation Details and Hyper-parameter Setup 2 (HPS2)

**Small datasets and EuroSAT RGB**  Here, we exactly reproduce the HP setup from (Brigato et al., 2022) including varying batch size, the Asynchronous Successive Halving Algorithm (ASHA) (Li et al., 2020a) with related parameters, and the repeated HP search over three different runs to ensure fair comparison against the benchmark. When fine-tuning VP+, the optimal configuration found during training is used to continue the training for the same epochs as before, only once for the optimal checkpoint.

### B.3  Implementation Details and Hyper-parameter Setup 3 (HPS3)

**OOD benchmarks on TinyImagenet**  We split the original Tiny Imagenet training set into 80%-20% to perform the HP selection for the default architecture. We run 200 trials each for 250 epochs and with a batch size of 128 samples. The learning rate and weight decay are respectively sampled randomly (log-uniformly) in $[10^{-4}, 10^0]$ and $[10^{-5}, 10^{-1}]$. We randomly sample also RandAugment strength (Cubuk et al., 2020) with N in $\{1, 2\}$ and M in $[5, 15]$. Furthermore, the parameters for MixUp and CutMix are randomly sampled from uniform distributions. Specifically, the Beta distribution parameter $\lambda_{\text{mu}}$ for MixUp is sampled uniformly from the range $[0.0, 1.5]$, while $\lambda_{\text{cm}}$ for CutMix follows the same uniform range $[0.0, 1.5]$. Additionally, the probability $p_{\text{mu}}$ of applying MixUp is uniformly sampled from the range $[0.0, 1.0]$.

## C  Additional Experimental Analyses

### C.1  Ablation on the characterization of tested distribution shifts

To better understand the tested distribution shifts from our experimental scenario, we follow the methodology proposed in (Ye et al., 2022), which classifies distribution shifts in *diversity* and *correlation* shifts.

We reproduced the setup available in the updated official code[2], including improvements regarding shift quantification stability. More precisely, a calibration step decreases the possibility of measuring a shift when the data is truly IID distributed. We employed an ImageNet pre-trained network for all datasets and used the default configuration regarding parameters. In the case of datasets with a resolution of $32{\times}32$, we substituted the original stem layer with a convolutional layer without the original aggressive stride and down-sampling required for $224{\times}224$ images.

---

[1] https://pytorch.org/vision/stable/transforms.html#torchvision.transforms.RandomResizedCrop

[2] https://github.com/m-Just/OoD-Bench

| Shift | c10 | ISIC | CUB | CLM | ESR | ESR |
|---|---|---|---|---|---|---|
| Diversity | $0.1390 \pm 0.0850$ | $0.0445 \pm 0.0246$ | $0.0100 \pm 0.0032$ | $0.0524 \pm 0.0243$ | $0.1597 \pm 0.2181$ | $0.2147 \pm 0.1103$ |
| Correlation | $0.035864 \pm 0.080210$ | $0.000016 \pm 0.000042$ | $0.006437 \pm 0.014169$ | $0.031068 \pm 0.031749$ | $0.013075 \pm 0.034594$ | $0.039201 \pm 0.081206$ |

| Shift | C10C | C100C | TINC | TINV2 | TINR | TINA |
|---|---|---|---|---|---|---|
| Diversity | $0.4427 \pm 0.0404$ | $0.3597 \pm 0.0655$ | $0.6490 \pm 0.0502$ | $0.0188 \pm 0.0076$ | $0.5106 \pm 0.0946$ | $0.0159 \pm 0.0045$ |
| Correlation | $0.076091 \pm 0.059526$ | $0.120628 \pm 0.031087$ | $0.015097 \pm 0.004866$ | $0.017754 \pm 0.019738$ | $0.053542 \pm 0.014585$ | $0.009135 \pm 0.019666$ |

Table 3: **Characterization of distribution shift per dataset.**

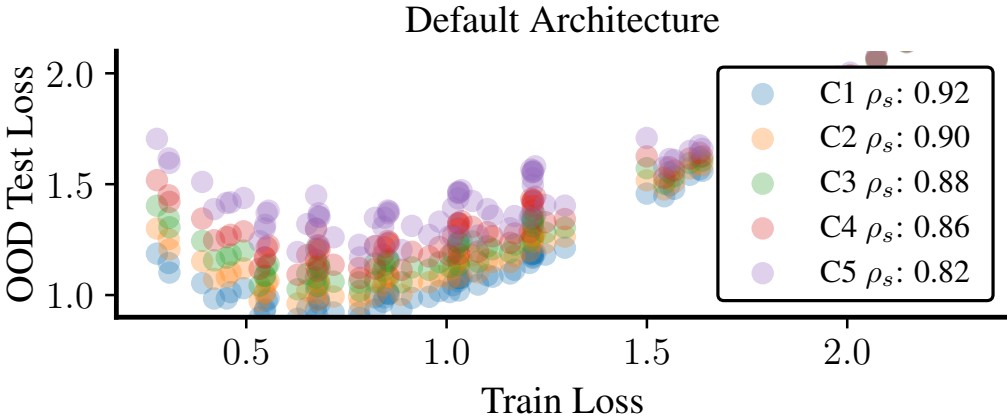

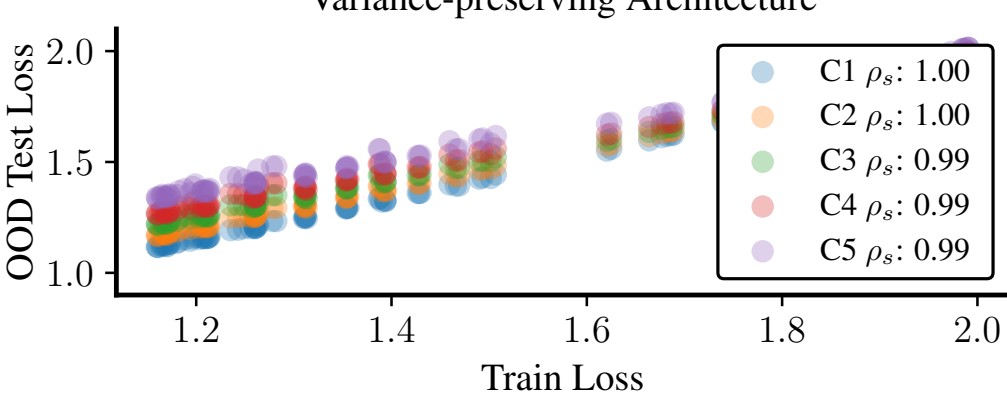

Figure 7: **Impact of distribution-shift strength.** The non-linear correlation remains almost unaffected for the variance-preserved MLP-4-2048, while it sharply drops for the default architecture.

We show the results of this analysis in Table 3. Means and standard deviations are computed across 8 repetitions. Interestingly, we find that all 12 tested datasets show OOD characteristics in mixed formats, including both diversity (more pronounced) and correlation (still present).

## C.2 ABLATION ON THE ROBUSTNESS OF THE ALIGNMENT FOR DISTRIBUTION SHIFT

In Figure 7, we ablate on the robustness of the alignment as the strength of the distribution shift increases through growing corruption levels on CIFAR10C for a grid search concerning $\alpha$ and $\lambda$ with the MLP-4-2048. For the default architecture, the strength of the monotonicity $\rho_s$ sharply decreases from 0.92 with the lowest corruption (C1) to 0.82 with C5. On the other hand, the VP remains barely unaffected, keeping almost perfect correlations (1.00 to 0.99), suggesting that it is more robust for handling distribution shifts.

