# OpenReview forum: "Can the Training Loss be Predictive for Out-of-Distribution Generalization?"
_ICLR.cc/2025/Conference — Submitted to ICLR 2025_

### Official Review · Reviewer_keeA · 2024-11-02

**Soundness:** 2
**Presentation:** 3
**Contribution:** 2
**Rating:** 3
**Confidence:** 3

**Summary:**

In this paper, the authors explore the research question: "whether the training loss can be used as a reliable predictor of OOD performance". The authors found that neural network architecture may be an important factor for the reliability of predictions. Then, the authors study the conditions and neural architectural designs to control the prediction-variance across distributions. Empirical results on example datasets demonstrate of the effectiveness of this methodology. There are several points the authors may address to further clarify this paper. See questions.

Overall, this is an interesting paper and some points can be addressed or even refocus on the HP optimization to further improve it. Current title seems to be too broad for this paper.

**Strengths:**

Good presentations and writtings.

**Weaknesses:**

See summary.

**Questions:**

1)	What are the potential practical implications for this paper? As this paper focus on the neural architectural design to improve the reliability for prediction. Then,

   1.1	does the proposed architecture brings additional benefits or drawbacks or equivalent performances compared with normal neural architectures.  The authors need to show the performance comparisons before and after architectural modifications.

   1.2	Have the authors explored any techniques to improve OOD performance prediction without modifying existing architectures? If not, what are the key challenges in doing so?"

2)	Comprehensive evaluations on different OOD distribution shifts. The authors are suggested to consider different kinds of OOD distribution shifts. As shown in [1] and [2], there are different types of distribution shifts in real-world datasets, it is suggested to consider two types of distribution shifts including diversity shifts and correlation shifts. In this paper, spurious correlations are not considered. Or alternatively, justifying why focusing on the diversity shift instead if this is the primary focus on this paper.

[1] A Critical Analysis of Out-of-Distribution Generalization
[2] Ood-bench: benchmarking and understanding out-of-distribution generalization datasets and algorithms

3) Experiments of this method for hyper-parameter optimization. As this method can potentially serve as a way to help improving HP optimization, the authors can consider introduce this to existing HP optimization pipelines to wider experiments, such as PACS, and ImageNet to see how it works. What advantages will this method demonstrate over other HP optimization baselines.  Besides, it could also (might) be beneficial if the focus of this paper be moved into the HP optimization as this shall have strong practical implications in various fields.

4) Experiments on standard OOD datasets. The authors could explain why choosing the datasets in this paper over standard OOD benchmark datasets. This paper can consider standard datasets in OOD generalization, such as PACS, ColoredMNIST,  VLCS, etc as well as what is done in most OOD generalization papers.

---

> ### Author Response · Authors · 2024-11-25
> **Rebuttal to Reviewer keeA**
>
> We thank the reviewer for the time taken to evaluate our paper. We address the questions in the following.
>
> **Q1.1. The authors need to show the performance comparisons before and after architectural modifications.**
>
> We would like to kindly point the reviewer to the main section of our paper (Section 2.2), in which we run an informative case study in which we dissect the impact of architectural modification across three different axes: single layers, depth, and width. In Figures 2-6, we show qualitative comparisons for each architectural modification.
>
> In Sections 3.2 and 3.3 (Tables 1 and 2), we show quantitative comparisons between the default and our architecture regarding predictability and generalization.
>
> **Q1.2. Have the authors explored any techniques to improve OOD performance prediction without modifying existing architectures? If not, what are the key challenges in doing so?"**
>
> In Section 2.2 of our paper, we explained the limitations that default architectures have, as explained by the growth of variance due to the current design of models.
>
> **Q2. Comprehensive evaluations on different OOD distribution shifts**
>
> We thank the reviewer for this suggestion, which we find insightful and also aligns with a request from reviewer U9Kt.
>
> We followed [1] to study the distribution shift of tested OOD datasets. We reproduced their setup following their updated [official code](https://github.com/m-Just/OoD-Bench), which includes improvements regarding the stability of shift quantification.
>
> Interestingly, we find that our tested 10 datasets all show OOD characteristics in also mixed formats, including both diversity (more pronounced) and correlation (still present), as visible in the following table:
>
>
> | Shift       | c10                     | ISIC                    | CUB                     | CLM                     | ESR                     | ESRC                    |
> | ----------- | ----------------------- | ----------------------- | ----------------------- | ----------------------- | ----------------------- | ----------------------- |
> | Diversity   | $0.1390 \pm 0.0850$     | $0.0445 \pm 0.0246$     | $0.0100 \pm 0.0032$     | $0.0524 \pm 0.0243$     | $0.1597 \pm 0.2181$     | $0.2147 \pm 0.1103$     |
> | Correlation | $0.035864 \pm 0.080210$ | $0.000016 \pm 0.000042$ | $0.006437 \pm 0.014169$ | $0.031068 \pm 0.031749$ | $0.013075 \pm 0.034594$ | $0.039201 \pm 0.081206$ |
>
> | Shift       | C10C                    | C100C                   | TINC                    | TINV2                   | TINR                    | TINA                    |
> | ----------- | ----------------------- | ----------------------- | ----------------------- | ----------------------- | ----------------------- | ----------------------- |
> | Diversity   | $0.4427 \pm 0.0404$     | $0.3597 \pm 0.0655$     | $0.6490 \pm 0.0502$     | $0.0188 \pm 0.0076$     | $0.5106 \pm 0.0946$     | $0.0159 \pm 0.0045$     |
> | Correlation | $0.076091 \pm 0.059526$ | $0.120628 \pm 0.031087$ | $0.015097 \pm 0.004866$ | $0.017754 \pm 0.019738$ | $0.053542 \pm 0.014585$ | $0.009135 \pm 0.019666$ |
>
> We will add the table in the Appendix to ensure the readers know the OOD characteristics of our experimental setup and refer to this analysis in the main paper.
>
> [1] OoD-Bench: Quantifying and Understanding Two Dimensions of Out-of-Distribution Generalization. CVPR 2021
>
> **Q3. Experiments of this method for hyper-parameter optimization. What advantages will this method demonstrate over other HP optimization baselines.**
>
> Unfortunately, we are not entirely sure what the reviewer refers to for “other HP optimization baselines”. Would it be possible to clarify?
>
> To provide a general overview, our method changes the model selection approach since it does not require validation data. As evidenced in Section 3.1 (Training setup), our approach can be used with both grid/random search and more advanced early-stopping HP optimization algorithms such as ASHA.
>
> In summary, we are not really sure how we could compare to other HP optimization baselines since our method is agnostic with respect to them. Instead, we compared against the standard model selection approach, i.e., train-validation split, which practically represents our baseline to compare against.

---

> > ### Author Response · Authors · 2024-11-25
> > **Rebuttal to Reviewer keeA (1)**
> >
> > **Q4. Experiments on standard OOD datasets.**
> >
> > We understand the popularity of datasets such as PACS, ColoredMNIST, and VLCS. However, we believe that we tested datasets that well represent the OOD literature as exemplified by the extensive use of domain shifts (e.g., -R, -A, and -V2 Imagenet versions) and robustness datasets (-C CIFAR/Imagenet versions) in seminal OOD works such as [1] and [2].
> >
> > We opted for small datasets representing a real-world OOD test that would enormously benefit from our approach (not requiring validation data). Then, we also tested domain shifts introduced by the ImageNet variants -R, -A, and -V2, but due to the high computational costs of studying model selection approaches at the Imagenet scale, we opted for the TinyImagenet versions [3]. Finally, we employed the robustness benchmarks representing real-world corruptions (as shown in [2]), representing a challenging and popular testbed for OOD generalization.
> >
> > [1] OoD-Bench: Quantifying and Understanding Two Dimensions of Out-of-Distribution Generalization, CVPR 2021
> >
> > [2] The Many Faces of Robustness: A Critical Analysis of Out-of-Distribution Generalization, ICCV 2021
> >
> > [3] Oehri et al., GenFormer -- Generated Images are All You Need to Improve Robustness of Transformers on Small Datasets, ICPR 2024

---

> ### Comment · Reviewer_keeA · 2024-11-25
> **Further comments**
>
> Thanks a lot for the updates of the results. Sorry for the confusion. HP optimization here means the hyper-parameter optimization. This method indeed indicates that when the neural network architecture is correctly modified, the test loss are more aligned with the training loss. This is possibly related to hyper-parameter optimization. Formulating it into a hyper-parameter optimization method may make this paper stronger as there are few papers in the hyper-parameter optimization area considering the generalization of hyper-parameter optimization.

---

> ### Author Response · Authors · 2024-11-25
>
> We thank the reviewer for providing additional clarifications regarding Q3.
>
> We agree that a big part of our work is about HP optimization because model selection implies searching for HPs. Indeed, we dedicated a Related Work section to it, named "Hyper-parameter tuning" and discussed how our approach is positioned in that landscape. Few works have tackled the problem of HP optimization without relying on validation sets.
> We discussed the limitations of such works there (see again the Related Work section).
>
> Therefore, considering our discussion on related work plus our experimental setting that implies the HP optimization for studying our research question (RQ), we believe that our work enriches the HP optimization literature in the current format. Does the reviewer agree on this point? We specifically focused on OOD problems because predicting generalization without needing OOD validation data, which is challenging to gather, gives our study an essential advantage and strengthens our contribution.
>
> We are available for additional clarifications if the reviewer finds them necessary and hope to have clarified the doubts raised during the first review period.

---

### Official Review · Reviewer_mCKs · 2024-11-02

**Soundness:** 2
**Presentation:** 3
**Contribution:** 2
**Rating:** 3
**Confidence:** 4

**Summary:**

This paper investigates whether training loss could be used to predict OOD performance. It claims that preserving consistent prediction variance across training and testing distributions is essential. This paper analyzes the factors that may affect prediction variance and further proposes variance-preserving architectures.

**Strengths:**

* The attempt to bridge the training loss with the model performance on the testing set, especially the OOD test set is interesting.

* Generally this paper is clear. The figures in this paper are pretty.

**Weaknesses:**

* The relationship between the training testing loss correlation and the correlation between each predicted label seems fra-fetched. As claimed in Eq.4, it requires the correlation between the predicted label of each of the two training testing data samples to be close to 1, which seems impossible.

* The experimental setting that random samples a subset from the CIFAR10 training set and tests on the CIFAR10 testing set does not qualify as an OOD scenario. The distribution of the CIFAR10 training set is the same as that of the CIFAR10 testing set. Randomly sampling a subset does not change the distribution.

* According to the results shown in Fig. 2, 3, 4, 5, and 6, these models show **the typical over-fitting pattern**. It seems that the proposed adjustment increases the correlation because it prevents over-fitting. However, overfitting is caused by the training on a small subset of the original training set, which could be simply dealt with.

Considering the above-mentioned weaknesses, I have doubts about the analyses in this paper and the proposed variance-preserving architecture. In my understanding, the proposed method limits the ability of the network to prevent overfitting, which also explains the performance drop as shown in Table 2.

**Questions:**

As mentioned in the weakness part, I have doubts about the analysis, the experimental setting, and the proposed method in this paper. I would like to see a rebuttal regarding the mentioned weaknesses. Currently, I will recommend reject for this paper.

---

> ### Author Response · Authors · 2024-11-15
> **Rebuttal to Reviewer mCKs**
>
> We would like to thank the reviewer for taking the time to read our submission and for valuable feedback concerning our paper. We discuss the three main weaknesses raised in the following.
>
> **W1. The relationship between the training testing loss correlation and the correlation between each predicted label seems fra-fetched. As claimed in Eq.4, it requires the correlation between the predicted label of each of the two training testing data samples to be close to 1, which seems impossible.**
>
> To address your concern effectively, we would appreciate clarifications on specific aspects regarding why the derived relationship seems far-fetched. Furthermore, our derivation is empirically validated in Section 2.2 and Section 3.
>
> As a general summary to clarify the point, we would like to refer the reviewer to Appendix A, where, as specified in Line 131, we provide all the details for the derivations of Equations (2) and (3), which are then critical to understanding Equation (4). We apply rules, such as the linearity property of the expectation and Taylor expansions of the loss function $\mathcal{L}$ around the mean prediction. Both variance and covariance are computed as a function of the hyper-parameters $\mathbf{h}$.
>
> Equation (4) shows that if model predictions $\hat{y} = f(\mathbf{x}, \mathbf{w}, \mathbf{h})$, with $\mathbf{h} \sim H$, i.e., over hyper-parameter space, maintain stability across distribution shifts $p(x) \rightarrow p^{\prime}(x)$ and behave similarly within each distribution, we can reliably align training and test losses. To maintain stability, we control the prediction variance via architecture design (Section 2.2). We empirically validate Equation (4) in the experimental section (Table 1).
>
> **W2. The experimental setting that random samples a subset from the CIFAR10 training set and tests on the CIFAR10 testing set does not qualify as an OOD scenario.**
>
> Learning from small/insufficient data has been cast as an OOD problem by [(Wad et al., 2022)](https://arxiv.org/pdf/2207.12258).
>
> Figure 2 from their work shows that subsampling a large dataset incurs the learner to face an OOD scenario for testing since the distribution to which it belongs is broader than the one from which the subsampled training set is drawn. Our aggressive sub-sampling (1%) for the CIFAR10 case study matches the same reasoning and qualifies as an OOD problem.
> On line 360, we referenced (Wad et al., 2022)  when introducing the experimented small-dataset scenarios. Still, we plan to reference it also on Line 197 when presenting the case study to avoid any source of confusion and improve clarity.
>
> Wad et al., Equivariance and Invariance Inductive Bias for Learning from Insufficient Data, ECCV 2022
>
> **W3. It seems that the proposed adjustment increases the correlation because it prevents over-fitting. However, overfitting is caused by the training on a small subset of the original training set, which could be simply dealt with.**
>
> We fully agree that our proposed adjustment increases the correlation because it prevents over-fitting.
>
> We respectfully disagree that overfitting is only caused by the training on a small subset of the original training set, as in practice, the picture is more complex given the following empirical evidence:
>
> - We show in Figure 7 (top) an example in which the overfitting pattern still happens for the default architecture even if the model was trained on the full CIFAR10 training set and tested on CIFAR10C. The reason for overfitting is not data scarcity but the distribution shift experienced when going from CIFAR10 (training) to CIFAR10C(1-5) (testing), plus some of the hyper-parameters from the search space that caused overfitting on the ID training dataset.
>
> - Our findings align with previous work that we referenced [(Teney et al., 2024)](https://arxiv.org/pdf/2209.00613), which did not experiment with any small datasets. Similar overfitting patterns (see Figure 2 of their paper) may only depend on the difference between ID and OOD data regardless of the size of the ID training dataset.

---

> ### Author Response · Authors · 2024-11-15
> **Rebuttal to Reviewer mCKs (1)**
>
> **W. In my understanding, the proposed method limits the ability of the network to prevent overfitting, which also explains the performance drop as shown in Table 2.**
>
> Our variance-preserving mechanism embedded in the architecture enables model selection directly from the training loss by preserving prediction variance and consequently preventing the model from overfitting the training set when extreme hyper-parameter configurations are tested and strong distribution shifts happen.
>
> This is a clear advantage that enables the predictability of generalization (Table 1). Dealing with overfitting when performing model selection with default architectures without validation data is challenging. See Figure 1 (top) and Table 1. We are unsure why the reviewer considers our architecture's “prevention of overfitting” as a weakness. We hope to have clarified this point.
>
> Finally, we agree that the constraints in the architecture seem to induce a trade-off in terms of performance (Table 2), but only for problems with many classes. We attribute this to the preservation of decorrelated representations and fixing of the BN affine parameters, as we explained in Lines 468-475, which becomes more challenging when you have many fine-grained classes to classify. We do not see that the broad prevention of overfitting causes this drop since there is no gap in problems with fewer classes.
>
>
> We hope to have clarified all doubts and are available for further clarification.

---

> ### Author Response · Authors · 2024-11-27
> **3rd Rebuttal to Reviewer mCKs**
>
> We thank the reviewer for providing additional clarification regarding his concerns. We provide additional explanations concerning the actual contribution of our work and the motivations behind it, which we hope will clarify the original concern of the reviewer. The initial concern can be synthesized by the following claim in the last comment:
>
> - **The key point is that various previous methods could prevent overfitting, and the proposed method is not desirable.**
>
> **P1. In the context of OOD generalization, we are unaware of any model-selection approach preventing overfitting that differs from splitting training data for validation**, e.g., see Section 3.1 of [In Search of Lost Domain Generalization](https://arxiv.org/pdf/2007.01434). In [(Miller et al., 2021)](https://proceedings.mlr.press/v139/miller21b/miller21b.pdf), authors indeed showed that IID test accuracy correlates with OOD test performance. Each point in the correlation plots represents a model trained with different HPs. If you optimize for IID test accuracy (== practically an IID validation set), then you predict OOD generalization well, which implies good model selection.
>
> **P2. Recent work [(Teney et al., 2024)](https://arxiv.org/pdf/2209.00613) interestingly found out that even when using IID validation data as a proxy for model selection, you may still get overfitting patterns** regardless of the training dataset size. These overfitting patterns (Figure 2 of [(Teney et al., 2024)](https://arxiv.org/pdf/2209.00613)) indeed align with what we empirically find (our figures).  **The reviewer agrees with this**:
>
> > good testing performance on one domain does not guarantee good performance on other domains.
>
> **We also do agree if you have a default architecture**, i.e., before our study.
>
> **Our RQ.** Following this empirical evidence, and motivated by the unreliability of IID validation data for OOD model selection, as discussed in **P2**, we asked ourselves if it is possible to more reliably predict OOD generalization without the need for any IID validation data.
>
> **Our contribution**. Our solution is a set of architectural modifications that prevent the variance from growing as we move from ID training data to OOD testing data. We indeed think that the key resides in how modern architectures are built. We show that our modifications enable reliable prediction of OOD generalization, including popular OOD benchmarks such as CIFAR10C/100C, TinyImagenet (versions -C/-A/-R/V2), and several small datasets (Table 1). **We achieve this without the need for any IID validation data.** Reviewer U9Kt also acknowledged our contribution and considered it a clear strength:
>
> > This is different from the traditional validation-based model selection and is particularly suitable for scenarios involving distribution shifts
>
> **What are the benefits of our approach?**
>
> 1. Going beyond the challenges of current model-selection paradigm relying upon validation which may arise in use cases where data is: 1) expensive, such as in medical imaging, 2) logistically unfeasible to collect, as in federated learning, 3) scarce hence inherently unreliable for unbiased evaluations, or 4) scarce hence unreliable when distribution shifts happen in real-world deployment (i.e., our discussed **P2**). We discussed these points in the Introduction.
> 2. Having a network that is stable across distribution shift, hence reliable, better calibrated etc.
>
> **With our solution, do we have a trade-off regarding absolute performance while achieving reliable OOD prediction?**
>
> We do, and we openly discussed it in Section 3.3. **We noticed that when the number of classes is small < 15, we reach equivalent accuracy to validation-based model selection, while when the number of classes rises, we observe a performance gap.** We provided intuitions on why this is happening in this particular case, both in this discussion thread and Section 3.3 of our paper.
>
> **However, our study does not tell the whole story since we are only the first to propose such a solution**, and we are sure that the community would be interested in understanding how we could further shoot this gap and investigate how/why this happens. E.g., longer training schedules managed to shoot the performance gap when the architecture was scale-invariant and enabled memory savings from the simpler optimizer as shown in [(Li et al, 2022)](https://proceedings.mlr.press/v162/li22b/li22b.pdf). For instance, in our CIFAR10C, we find that, indeed, the additional fine-tuning (i.e., VP vs VP+) raised the performance from 58.8% to 60.8% (Table 1).
>
> We hope that this general summary, adapted to better explain the concerns regarding the reviewer's raised weaknesses, provide a better evidence on our motivations and contributions.

---

> > ### Comment · Reviewer_mCKs · 2024-12-02
> >
> > Thanks for the authors' detailed rebuttal. I truly appreciate the author's rebuttal. However, I regret to say that my concerns are not resolved. Below are my concerns and responses to the authors.
> >
> > **Methods Regarding Overfitting**
> >
> > > The authors said  *"In the context of OOD generalization, we are unaware of any model-selection approach preventing overfitting that differs from splitting training data for validation"*
> >
> > * **First of all, various methods can be used to prevent overfitting without a validation set, e.g., cross-validation (which can be found in textbooks about statistic learning), boosting [1], and bootstrap aggregating [2].** If the authors want to present the proposed method as a model-selection approach preventing overfitting, experiments comparing the proposed method with other baseline methods should be provided.
> >
> > * Secondly, given the much-degraded accuracy of the proposed architecture, I can not help but wonder whether splitting a small subset of training data as a validation set would be a better solution. This paper does not provide enough experimental results to support the effectiveness of the proposed method.
> >
> > **Concerns regarding OOD**
> >
> > The term "out of distribution (OoD)" repeatedly appears in the title of the paper, the main content of the paper, and in the authors' rebuttal. However, the proposed method does not appear to have a strong connection with the OOD scenario.
> >
> > * According to the latest response provided by the authors, [3] showed that IID test accuracy correlates with OOD test performance. It indicates that one can correlate training loss with OOD test performance by correlating training loss with IID test accuracy which makes the proposed method trivial.
> >
> > * Generally, OOD generalization focuses on generalizing models to unseen data domains. However, the proposed method is motivated by **tackling the problem of limited IID validation data in OOD scenarios** which is weird. I honestly doubt if it is novel or meaningful to the OOD generalization area.
> >
> > After the discussion with the authors, despite the authors providing extensive responses, my concerns are not resolved. I appreciate the effort made by the authors but I will keep my rating.
> >
> > [1] Freund Y. Boosting a weak learning algorithm by majority[J]. Information and Computation, 1995, 121(2): 256-285.
> >
> > [2] Breiman L. Bagging predictors[J]. Machine learning, 1996, 24: 123-140.
> >
> > [3] Miller J P, Taori R, Raghunathan A, et al. Accuracy on the line: on the strong correlation between out-of-distribution and in-distribution generalization[C]//International conference on machine learning. PMLR, 2021: 7721-7735.

---

> ### Author Response · Authors · 2024-12-03
> **4th Rebuttal to Reviewer mCKs**
>
> We thank the reviewer for the additional response. Again, we would like to provide our perspective on some points we respectfully disagree with and some additional clarifications.
>
> - **First of all, various methods can be used to prevent overfitting without a validation set, e.g., cross-validation**
>
> Cross-validation repeats the splitting procedure k- times. This approach is conceptually the same as our baseline, which is just less computationally heavy, given the cost of multi-split training in deep learning.
>
> - **If the authors want to present the proposed method as a model-selection approach preventing overfitting, experiments comparing the proposed method with other baseline methods should be provided.**
>
> In the previous comment, we referenced a **popular benchmark in the OOD community** ([In Search of Lost Domain Generalization](https://arxiv.org/pdf/2007.01434)) which **recommended the use of three model selection approaches**: **1) Training-domain validation set**, 2) Leave-one-domain-out cross-validation, 3) Test-domain validation set. Please note that both 1) and 2) split training data for validation. **Precisely, our baseline corresponds to 1).** 2) is only applicable if you have environmental labels (less general), and 3) is an unrealistic upper bound for OOD generalization.
>
> - **Secondly, given the much-degraded accuracy of the proposed architecture, I can not help but wonder whether splitting a small subset of training data as a validation set would be a better solution.**
>
> We openly discussed this trade-off in Section 3.3. **We noticed that when the number of classes is small < 15, we reach equivalent accuracy to validation-based model selection, while when the number of classes rises, we observe a performance gap.** We provided intuitions on why this is happening in this particular case, both in this discussion thread and Section 3.3 of our paper.
>
> - **According to the latest response provided by the authors, [3] showed that IID test accuracy correlates with OOD test performance.**
>
> We also clarified in point P2 of the previous answer that a recent study (Teney et al., 2024) showed that the findings from [3] do not always hold in practice, making IID validation performance unreliable in some cases.
>
> - **However, the proposed method is motivated by tackling the problem of limited IID validation data in OOD scenarios which is weird.**
>
> We clarify that, by definition, an **OOD problem does not have available validation data, which makes our method of predicting OOD generalization from the IID training data very suitable.** Reviewer U9Kt also acknowledged this fact:
>
> > This is different from the traditional validation-based model selection and is particularly suitable for scenarios involving distribution shifts.

---

### Official Review · Reviewer_9YgD · 2024-11-02

**Soundness:** 2
**Presentation:** 2
**Contribution:** 2
**Rating:** 3
**Confidence:** 4

**Summary:**

This work investigates the conditions under which the training loss can predict OOD generalization. The authors find that maintaining consistent prediction variance across training and testing distributions is key to correlating training loss with OOD generalization. They propose architectural adjustments to preserve this variance and demonstrate that these variance-preserving architectures allow for reliable model selection based on training loss in various OOD scenarios.

**Strengths:**

1. The study of OoD generalization and model robustness has always been an important topic. Investigating model selection and generalization ability from the perspective of network architecture is a worthwhile endeavor.

2. The paper is well written and easy to follow.

**Weaknesses:**

1. The authors’ claim that the RQ in line 69 is an unexplored research question is inaccurate, as this topic has already been studied extensively in prior work, such as [1] and [2]. This study's approach, which examines the problem from the perspectives of variance preservation and network architecture, is also not new; similar analyses have been conducted in previous studies, such as [3] and [4]. This work does not sufficiently compare itself to these prior studies and build upon them for further exploration.

2. It is inappropriate to directly ignore the gradient term $\nabla_f \mathcal{L}$ in formulas (2) and (3). It is a function of $𝑓$ and $\boldsymbol{w}$, and choosing different $𝑓$ and $\boldsymbol{w}$ will yield different $\nabla_f \mathcal{L}$, thereby impacting $\text{Cov}(J(\boldsymbol{w}, \boldsymbol{h}), J'(\boldsymbol{w}, \boldsymbol{h}))$ and $\text{Var}(J(\boldsymbol{w}, \boldsymbol{h}))$. Considering only the Cov term in formulas (2) and (3) leads to an incomplete conclusion.

3. The architectural adjustments proposed in this work are already commonly used for training neural networks and do not provide any new insights.

[1] Ishaan Gulrajani, David Lopez-Paz, In Search of Lost Domain Generalization, ICLR 2021.

[2] Ye et al., Towards a Theoretical Framework of Out-of-Distribution Generalization, NeurIPS 2021.

[3] Krueger et al., Out-of-Distribution Generalization via Risk Extrapolation, ICML 2021.

[4] Bai et al., NAS-OoD: Neural Architecture Search for Out-of-Distribution Generalization, ICCV 2021.

**Questions:**

1. With a substantial amount of research already available, why is the RQ in line 69 considered an unexplored research question? Is there a key point I might have overlooked?

2. Why can the gradient term $\nabla_f \mathcal{L}$ in formulas (2) and (3) be ignored?

3. Which of the architectural adjustments proposed in this work are newly introduced?

Please see Weaknesses for details.

---

> ### Author Response · Authors · 2024-11-15
> **Rebuttal to Reviewer 9YgD**
>
> We would like to thank the reviewer for taking the time to read our submission and for valuable feedback concerning our paper.
>
> **Q1. With a substantial amount of research already available, why is the RQ in line 69 considered an unexplored research question? Is there a key point I might have overlooked?**
>
> Our RQ is unexplored due to a fundamental distinction in model selection (MS) approaches. The referenced previous work [1,2,3,4] needs a $validation$ signal to run MS. Instead, we solely rely on the training loss (ERM). We further explain this fundamental difference in detail in the following.
>
> In **[1]** (Section 3.1), the authors propose three ways to perform MS, all of which depend on a validation signal:
> - MS1: Training-domain validation set
> - MS2: Leave-one-domain-out cross-validation
> - MS3: Test-domain validation set (oracle, unrealistic)
>
> **Our approach introduces a fourth unexplored strategy: training on all domains and performing MS using only the training loss (ERM).** Note that in our paper, we are comparing against MS1.
>
> Similarly, in the Introduction of **[2]**, the authors state: “Inspired by our bounds, we propose a model selection method to select models with high validation accuracy and low variation”.
> Therefore, the authors employ the validation accuracy coming from either MS1 or MS2.
>
> Finally, the authors employ validation signals in **[3]** and **[4]**.
> In [3], the hyper-parameters $\lambda_{min}$ from MM-REx (Equation 6) and $\beta$ from V-REx must be selected from validation data. Indeed, in Section 4.3, the authors state that they evaluate DomainBed [1] using MS1.
> In [4], the validation signal to perform NAS, $l_{val}$ in their Equation (4), comes from generated unseen domain data. In practice, the approach synthesizes an unseen validation set by taking advantage of generative models and the assumption of having environmental labels. However, the principle of doing MS with respect to an unknown $validation$ signal holds and is fundamentally different from what we proposed. Indeed, in Section 3.1, the authors stress the need for diverse validation data to guide NAS, which is what we are challenging with our MS strategy from the training loss alone.
>
> We thank the reviewer for highlighting these works, which we plan to include in the related work section to clarify distinctions and better emphasize the originality of our contribution.
>
> **Q2. Why can the gradient term $\nabla_f \mathcal{L}$ in formulas (2) and (3) be ignored?**
>
> The $\nabla_f \mathcal{L}$ can be ignored because it represents a constant that scales the contribution of the Cov and Var terms, and hence, it simply acts as a scalar weight.
> To further clarify, we would like to point the reviewer to the full derivation of $\textit{Cov}(J(\mathbf{w}, \mathbf{h}), J'(\mathbf{w}, \mathbf{h}))$ in Appendix A and make an analogy with a simplified setting.
>
> If we perform a first-order Taylor expansion of function $g(x)$ around a point $a$, we get $g(x) \approx g(a) + g’(a) (x-a)$. Note that if we compute the expectation of the products of the first-order terms around two points $a$ and $b$, we get $E_{x}[g’(a) g’(b) (x-a) (x-b)]$. Both $g’(a)$ and $g’(b)$ can be taken out from $E_{x}$ because they are independent of $x$ and act as two constants multiplying the covariance term $E_{x}[(x-a)(x-b)]$, which plays the central role.
>
> We perform a similar expansion of the loss around the mean prediction $\mathbf{\mu}$ (see Equations (8) or (13)). The derivative $g’$ corresponds to $\nabla_f \mathcal{L}$, point $a$ to $\mathbf{\mu_i} = E_{\mathbf{h}}(f(\mathbf{x_i}), \mathbf{w}, \mathbf{h})$, and point $b$ to $\mathbf{\mu_j} = E_{\mathbf{h}}(f(\mathbf{x_j}), \mathbf{w}, \mathbf{h})$. Note that our points correspond to the average prediction over the space of hyper-parameters $\mathbf{h}$. Hence $\nabla_f \mathcal{L}(y_i, \mathbf{\mu_i})$ and $\nabla_f \mathcal{L}(y_j, \mathbf{\mu_j}))$ are scalars.
>
> As a side note, we are not considering the target y in the simplified example since it is a constant. Moreover, the $\nabla_f \mathcal{L}$ terms can be taken out of $E_{\mathbf{h}}$ because $\mathbf{\mu_i}$|$\mathbf{\mu_j}$ is defined as an expectation over $\mathbf{h}$ which removes any dependency on the randomness of $\mathbf{h}$ at the level of $\mathbf{\mu_i}$|$\mathbf{\mu_j}$.

---

> > ### Comment · Reviewer_9YgD · 2024-11-23
> >
> > For Q1:
> > There are many works that use training loss to study the OoD generalization ability of deep learning models. Arjovsky et al. ([5]) try to minimize training loss in each training environment while ensuring that the learned representations remain consistent. Sagawa et al. ([6]) analyzes the issues inherent in worst-case training loss minimization for OoD generalization. If the authors believe that previous works have issues, they can point them out, propose new solutions, and compare their approach with the prior works. However, this comparison is missing in the current study.
> >
> > [5] Arjovsky et al., Invariant Risk Minimization, 2019.
> >
> > [6] Sagawa et al., Distributionally Robust Neural Networks for Group Shifts: On the Importance of Regularization for Worst-Case Generalization, ICLR 2020.
> >
> > For Q2: According to the authors' explanation, $𝑓$ and $\boldsymbol{w}$ are fixed in $\boldsymbol{\mu}_i$ in line 880. Then, taking the expectation with respect to $\boldsymbol{h}$, $(f(\boldsymbol{x}_i, \boldsymbol{w}, \boldsymbol{h}) - \boldsymbol{\mu}_i)$ does not vanish since $𝑓$ and $\boldsymbol{w}$ are variables in the first term. Hence, formula (9) is not correct. Similarly, formula (14) is not correct.

---

> ### Author Response · Authors · 2024-11-15
> **Rebuttal to Reviewer 9YgD (1)**
>
> **Q3. Which of the architectural adjustments proposed in this work are newly introduced?**
>
> We would like to clarify this point further, which we think is crucial to fully understanding the contribution our work.
>
> **Our novel contribution is an architecture that preserves variance and consequently enables model selection from the training loss alone. An architecture that fulfills our desiderata does not exist in the literature.**
>
> For this reason, we applied adjustments that come from different components, yet with several technical difficulties (e.g., see Section 2.3) that would not fulfill our goal of variance preservation if taken individually, as we explained in Section 2.2. There could be many ways to preserve variance in existing/future architectures, e.g., employing densely connected residuals, as done in DenseNets, instead of additive residuals, as in ResNets, for controlling variance growth along depth.
>
> We are providing the community with a possible solution with the intent of further exploration. The novelty comes from the variance preservation as a whole, not in the single components.

---

> ### Author Response · Authors · 2024-11-24
>
> We thank the reviewer for engaging in the discussion. We still think that there are some misunderstandings regarding both Q1 and Q2, so we add additional evidence in the following.
>
> **Q1. There are many works that use training loss to study the OoD generalization ability of deep learning models.**
>
> We would kindly point out that we never claimed the absence of existing works that use the training loss to study the OoD generalization ability of DL models.
> **We claim that our RQ is novel because none of the previous work, including [1,2,3,4,5,6], studied the predictability of OoD performance when performing model selection from only the training loss.**
>
> Similarly to [1,2,3,4], **[5]** and **[6]** also belong to the model selection (MS) categories of MS1 or MS2 (previous comment). Hence, they depend again on external $\textit{validation}$ signal for model selection. For instance, the IRM objective from [5] necessitates a validation set to tune parameter $\lambda$ defined in Equation (1). The authors indeed state that: "$\lambda \in [0, \inf)$  is a hyper-parameter balancing predictive power and invariance.". **The authors did not study if/how $L_{IRM}$ aligns with $L_{test}$ as $\lambda$ varies (one of many possible HPs), i.e., what we analyze in our novel RQ.**
>
> We do not believe that such previous works have issues but rather that they study a different problem. We find our work more related to the referenced works from our paper by [(Miller et al., 2021)](https://proceedings.mlr.press/v139/miller21b/miller21b.pdf) and [(Teney et al., 2024)](https://arxiv.org/pdf/2209.00613). However, we still believe that discussing the differences between [1,2,3,4,5,6] and our paper better emphasizes the originality of our contribution.
>
> We hope to have clarified the misunderstanding regarding our RQ and are available for more discussion if the reviewer finds it necessary.
>
> Miller et al., Accuracy on the line: on the strong correlation between out-of-distribution and in-distribution generalization, 2021
> Teney et al., Id and ood performance are sometimes inversely correlated on real-world datasets, 2024
>
> **Q2. Hence, formula (9) is not correct. Similarly, formula (14) is not correct.**
>
> We believe that both formulas (9) and (14) are correct.
>
> To go from (8) to (9), we apply $E[x + a] = E[x] + E[a]$, $E[E[x]] = E[x]$, and the definition $E[x] = a$:
>
> $$
> E_{\textbf{h}}({f(\mathbf{x_i}, \mathbf{w}, \mathbf{h})} - \mu_i) = E_{\textbf{h}}({f(\mathbf{x_i}, \mathbf{w}, \mathbf{h})}) - E_{\textbf{h}}(\mu_i) = E_{\textbf{h}}({f(\mathbf{x_i}, \mathbf{w}, \mathbf{h})}) - E_{\textbf{h}}(E_{\textbf{h}}({f(\mathbf{x_i}, \mathbf{w}, \mathbf{h})})) = E_{\textbf{h}}({f(\mathbf{x_i}, \mathbf{w}, \mathbf{h})}) - E_{\textbf{h}}({f(\mathbf{x_i}, \mathbf{w}, \mathbf{h})}) = 0
> $$

---

> > ### Comment · Reviewer_9YgD · 2024-11-25
> >
> > For Q2: According to the author's explanation, $𝑓$ and $\boldsymbol{w}$ are fixed in the definition of $\boldsymbol{\mu}_i = E_h (f(\boldsymbol{x}_i, \boldsymbol{w}, \boldsymbol{h}))$, i.e., $\boldsymbol{\mu}_i$ is a constant. However, $𝑓$ and $\boldsymbol{w}$ are variables in the first term of $f(\boldsymbol{x}_i, \boldsymbol{w}, \boldsymbol{h}) - \boldsymbol{\mu}_i$. Hence, taking the expectation with respect to $\boldsymbol{h}$, $E_h (f(\boldsymbol{x}_i, \boldsymbol{w}, \boldsymbol{h}))$ is not equal to $\boldsymbol{\mu}_i$.

---

> > > ### Author Response · Authors · 2024-11-25
> > >
> > > Our assumption is that each $\mathbf{h} \sim \mathcal{H}$ defines a specific $\mathbf{w}$ which in turn defines a specific $f$.
> > >
> > > In other words, we do not take into consideration the randomness deriving from each learning trajectory given a fixed $\mathbf{h}$. We need this simplification to keep the derivation more tractable. In practice, we think it is reasonable to assume that for a fixed HP setting, the learned prediction function $f(\cdot, \mathbf{w})$ does not vary meaningfully. E.g., if we fix $\mathbf{h}$ = (LR=0.1, WD=0.1), you will approximately obtain the same function $f(\cdot, \mathbf{w})$. In real-world experiments, the variation is indeed negligible when fixing HP settings. More importantly, the implications derived from Equation (4) do empirically hold in practice, as we showed in Section 2.2. and 3.
> > >
> > > To address the reviewer's concern, we can update the notation from $f(\mathbf{x_i}, \mathbf{w}, \mathbf{h})$ to $f_{\mathbf{h}}(\mathbf{x_i})$ and specify explicitly the assumption that we make. Therefore, the empirical computation of $E_{\mathbf{h}}[f_{\mathbf{h}}(\mathbf{x_i})]$ corresponds to an average over the HP-dependent functions $f_{\mathbf{h}}(\mathbf{x_i})$, given a specific point $\mathbf{x_i}$. We would like to keep the $f$, and not use only $\mathbf{h}(x_i)$ to cummincate that there is a learned function under the hood.

---

> > > > ### Comment · Reviewer_9YgD · 2024-11-26
> > > >
> > > > If each $\boldsymbol{h}$ defines a specific $\boldsymbol{w}$ which in turn defines a specific $f$, then the theoretical analysis in Section 2.1 loses its significance. Based on the content of the main text, $f$ represents the neural architecture. The conclusion that the theoretical analysis in Section 2.1 aims to demonstrate is that if a "good" neural architecture can be found, then the training loss and test loss can exhibit strong correlation across different hyperparameters $\boldsymbol{h}$. Conversely, for a "bad" neural architecture, whether the training loss can predict the test loss will be influenced by the choice of hyperparameters $\boldsymbol{h}$. Figure 1 is intended to illustrate this conclusion. Specifically, the Pearson correlation $\rho_{J \rightarrow J'}$ in Definition (1) is a function of $𝑓$ and $\boldsymbol{w}$, which would be better written as $\rho_{J \rightarrow J'} (𝑓, \boldsymbol{w})$. This theoretical analysis aims to demonstrate that a "good" neural architecture $f$ can make $\rho_{J \rightarrow J'} (𝑓, \boldsymbol{w})$ closer to 1.
> > > >
> > > > Actually, the influence of $\boldsymbol{w}$ on $\rho_{J \rightarrow J'} (𝑓, \boldsymbol{w})$ also needs to be analyzed. In fact, $\boldsymbol{w}$ represents the optimization method. Given $𝑓$ and $\boldsymbol{h}$, different optimization methods will result in different $\boldsymbol{w}$. A "good" optimization method can also make $\rho_{J \rightarrow J'} (𝑓, \boldsymbol{w})$ closer to 1. For example, Foret et al. ([7]) proposes sharpness-aware minimization to reduce the gap between the training loss and the test loss.
> > > >
> > > > Returning to the initial question "Why can the gradient term $\nabla_f \mathcal{L}$ in formulas (2) and (3) be ignored", $\nabla_f \mathcal{L}$ is a function of $𝑓$ and $\boldsymbol{w}$, representing the prediction accuracy of $𝑓$ and $\boldsymbol{w}$. For an OoD problem, we need to consider two factors: prediction accuracy and stability, as highlighted in many previous works. Considering only stability is insufficient. An extreme example would be choosing $𝑓$ and $\boldsymbol{w}$ that predict all samples as the same label. While this is highly stable, it is meaningless. Focusing solely on stability leads to the performance drop observed in Table 2.
> > > >
> > > > I appreciate the effort the authors have put into this work, but I still think that the current version has not reached the level of acceptance.
> > > >
> > > > [7] Foret et al., Sharpness-Aware Minimization for Efficiently Improving Generalization, ICLR 2021.

---

> ### Author Response · Authors · 2024-11-26
>
> We thank the reviewer for diving deeper into our paper, and we confirm that our "theoretical analysis aims to demonstrate that a "good" neural architecture $f$ can make $\rho_{J \rightarrow J^{\prime}}$ closer to 1". We kindly request the reviewer to consider our perspective on the following points:
>
> - **Actually the influence of $\mathbf{w}$ on $\rho_{J \rightarrow J^{\prime}}$ also need to be analyzed. In fact, $\mathbf{w}$ represents the optimization method.**
>
> We fully agree that the optimization algorithm significantly impacts the learning process. **Since our goal was to study the impact of the architecture $f$ alone (see Abstract), we fixed the optimizer to SGD as specified in Lines 190-191.**
>
> Several possible approaches exist that focus on predicting generalization, and indeed, we reported several of them in our *Predicting generalization* paragraph in the Related Work section. SAM, referenced in our paper, belongs to the category of using the proxy of wider minima for better generalization. Yet, the SAM paper did not study the prediction of generalization given different HPs. More precisely, if they focused on aligning train and test losses from the optimizer's perspective, they still did not study if/how the optimizer can maintain the alignment as $\mathbf{h}$ varies (e.g., LR/WD, etc.). This is once again a difference in contribution.
>
> **Reasonably, we believe that the topic is broad enough that a comprehensive analysis of the optimizer** ($\mathbf{w}$), along with all related choices and experiments, **extends far beyond the intended scope of our paper.**
>
> - **we need to consider two factors: prediction accuracy and stability ... Considering only stability is insufficient.**
>
> There is definitely some trade-off between prediction accuracy and stability. As we discussed in Section 3.3, we only noticed clearly worse performance for problems with many classes. We attribute this to the preservation of decorrelated representations and fixing of the BN affine parameters, which makes classification more challenging and may require more extended learning when you have many fine-grained classes to recognize. Connected to this, longer schedules may possibly shoot the gap with original architectures (Section 3.3):
>
> > In line with observations from previous studies, a further fine-tuning step might be needed because SI architectures necessitate slightly longer training schedules to reach the same performance due to the inherent constraints within the network architecture (Li et al., 2022).
>
> Generally, additional research may further improve/fill this gap.
>
>  **Regarding the "sole" consideration of stability, which is only partially true given we learn on training data, it leads to prediction results that are on par with default architectures on several datasets (Table 2) while at the same time extremely stable across broad experimental/HP settings (Table 1).** **We** also **stress that** such **prediction of generalization** is achieved **without** using any **validation** signal - which represents an important **novelty compared to all previous research referenced by the reviewer** that we plan to include for a broader paper perspective [1,2,3,4,5,6].
>
> Our results are indeed valuable for our targeted application domain, where validation data is scarce or prone to OOD shifts, since I can potentially optimize my HPs without needing to search for environmental data from different environments. **The reviewer mainly focuses on prediction accuracy. Still, equally importantly, we find that stable architectures may have practical importance in high-risk environments where stability is more relevant than prediction accuracy alone**.

---

### Official Review · Reviewer_U9Kt · 2024-11-04

**Soundness:** 3
**Presentation:** 3
**Contribution:** 3
**Rating:** 6
**Confidence:** 3

**Summary:**

This work investigates the use of training loss to predict the generalization performance of neural networks on out-of-distribution scenarios, instead of using the validation data. This work is particularly useful in distribution-shifted scenarios, where standard validation data may be unavailable. This work examines the conditions for establishing linear relationships between training and test losses. Comprehensive empirical analysis is provided to analyze the model-selection capabilities of the proposed framework through extensive experiments setup, including optimizing several hyper-parameters across popular OOD benchmarks.

**Strengths:**

1. This work proposes a novel approach for model selection by relying solely on training loss to predict out-of-distribution generalization. This is different from the traditional validation-based model selection and is particularly suitable for scenarios involving distribution shifts.

2. This paper analyzes the conditions for which training loss can correlate with OOD generalization, and develop a variance-preserving architecture based on these conditions.

3. Various hyper-parameters are thoroughly investigated across different types of OOD benchmarks, including corruption and domain shifts.

**Weaknesses:**

1. It would be valuable to analyze and investigate on different types of OOD shifts, including diversity shifts and correlation shifts [1], and discuss whether the proposed conditions hold across these two dimensions of OOD shifts.

2. Extending the evaluation to more complex real-world natural images, such as the WILDS benchmarks [2].

3. Providing additional theoretical justification for supporting and explain the proposed variance preserving architecture, would enhancing the reasoning and foundation of the proposed framework.

Reference:

[1] OoD-Bench: Quantifying and Understanding Two Dimensions of Out-of-Distribution Generalization. CVPR 2021.

[2] WILDS: A Benchmark of in-the-Wild Distribution Shifts. ICML 2021.

**Questions:**

Refer to the detailed suggestions provided in the weaknesses section.

---

> ### Author Response · Authors · 2024-11-25
> **Rebuttal to Reviewer U9Kt**
>
> We thank the reviewer for taking the time to read our submission and for valuable feedback concerning our paper. We particularly appreciate the positive overall response to our work. We reply to the weaknesses in the following.
>
> **W1. It would be valuable to analyze and investigate on different types of OOD shifts**
>
> We thank the reviewer for this suggestion, which we find insightful and also aligns with a request from reviewer keeA.
>
> We followed [1] to study the distribution shift of tested OOD datasets. We reproduced their setup following their updated [official code](https://github.com/m-Just/OoD-Bench), which includes improvements regarding the stability of shift quantification.
>
> Interestingly, we find that our tested 12 datasets all show OOD characteristics in also mixed formats, including both diversity (more pronounced) and correlation (still present), as visible in the following table:
>
> | Shift       | c10                     | ISIC                    | CUB                     | CLM                     | ESR                     | ESRC                    |
> | ----------- | ----------------------- | ----------------------- | ----------------------- | ----------------------- | ----------------------- | ----------------------- |
> | Diversity   | $0.1390 \pm 0.0850$     | $0.0445 \pm 0.0246$     | $0.0100 \pm 0.0032$     | $0.0524 \pm 0.0243$     | $0.1597 \pm 0.2181$     | $0.2147 \pm 0.1103$     |
> | Correlation | $0.035864 \pm 0.080210$ | $0.000016 \pm 0.000042$ | $0.006437 \pm 0.014169$ | $0.031068 \pm 0.031749$ | $0.013075 \pm 0.034594$ | $0.039201 \pm 0.081206$ |
>
> | Shift       | C10C                    | C100C                   | TINC                    | TINV2                   | TINR                    | TINA                    |
> | ----------- | ----------------------- | ----------------------- | ----------------------- | ----------------------- | ----------------------- | ----------------------- |
> | Diversity   | $0.4427 \pm 0.0404$     | $0.3597 \pm 0.0655$     | $0.6490 \pm 0.0502$     | $0.0188 \pm 0.0076$     | $0.5106 \pm 0.0946$     | $0.0159 \pm 0.0045$     |
> | Correlation | $0.076091 \pm 0.059526$ | $0.120628 \pm 0.031087$ | $0.015097 \pm 0.004866$ | $0.017754 \pm 0.019738$ | $0.053542 \pm 0.014585$ | $0.009135 \pm 0.019666$ |
>
> We will add the table in the Appendix to ensure the readers know the OOD characteristics of our experimental setup and refer to this analysis in the main paper.
>
> [1] OoD-Bench: Quantifying and Understanding Two Dimensions of Out-of-Distribution Generalization. CVPR 2021
>
>
> **W2. Extending the evaluation to more complex real-world natural images, such as the WILDS benchmarks**
>
> While we understand the importance of the WILDS benchmark, we would like to point out that we did our best to include in our experimental collection datasets encompassing real-world images coming from, e.g., medical/satellite//hand-written domains.
>
> For instance, our sub-sampled ISIC2018 dataset (ISIC in Tables 1 and 2) was originally part of a [biomedical imaging challenge](https://arxiv.org/pdf/1902.03368) concerning skin-lesion classification hosted at MICCAI 2018.
> In our opinion, the complexity/real-world nature of our ISIC aligns, for instance, with Camelyon17 (histopathology tumor images) present in the WILDS benchmark.
>
> We acknowledge the difference scale in terms of overall training images. Unfortunately,  it would have been computationally prohibitive to run large HP searches (100/200 trials), such as the ones proposed in our work, on datasets with 100x more training samples, such as Camelyon17.
>
> **W3. Providing additional theoretical justification for supporting and explain the proposed variance preserving architecture, would enhancing the reasoning and foundation of the proposed framework.**
>
> We agree on the importance of theoretical justification concerning our variance-preserving architecture. Would it be possible to be more precise regarding what the reviewer is actually interested in?
>
> In Section 2.1 and corresponding Appendix A, we analyzed the conditions for train-test loss alignment, which we think is the most "theoretical" part of our work.
>
> In Section 2.2, we focused on a running example to study how the different architectural components influence prediction variance.

---

> > ### Comment · Reviewer_U9Kt · 2024-12-03
> >
> > Thank you to the authors for the detailed response. Most of my original concerns have been addressed; however, I agree with the overfitting issues raised by Reviewer mCKs. I will keep my original rating.

---

> > > ### Author Response · Authors · 2024-12-03
> > > **2nd Rebuttal to Reviewer U9Kt**
> > >
> > > We thank the reviewer for replying to our rebuttal, and **we appreciate that the effort we put in the rebuttal addressed most of the original reviewer's concerns.**
> > >
> > > Unfortunately, we tend to respectfully disagree with the overfitting issues raised by Reviewer mCKs, as we thoroughly discussed during the rebuttal period in the related thread.
> > >
> > > Indeed, even a standard model selection pipeline, such as a train-validation split (cross-validation corresponds to the repetition of this procedure k times, but the concept is exactly the same), may suffer from overfitting due to some specific HP configurations that learn underspecified features for OOD generalization. This fact is visible in [(Teney et al., 2024)](https://arxiv.org/pdf/2209.00613), Figure 2 of their work.
> > >
> > > The whole point of our work is to prevent overfitting, such as these patterns do not appear, and you can reliably predict generalizing configurations from the training loss alone.

---

### Comment · Area_Chair_rDRX · 2024-12-01

Dear Reviewers,

If you have not already responded to the latest comments by the authors, please engage with them to clarify any remaining issues as the discussion period is coming to a close in less than a day (2nd Dec AoE for reviewer responses).

Thanks for your service to ICLR 2025.

Best,
AC

---

### Author Response · Authors · 2024-12-04
**Summary of Rebuttal period**

We thank all the reviewers for the time they invested in evaluating our paper and for their constructive feedback.  Below, we summarize the strengths, our revisions, clarifications, and points of disagreement.

# Strengths

**S1.** Reviewer U9Kt considered our approach novel and particularly suitable for scenarios involving distribution shifts.

**S2.** Reviewer 9YgD appreciated the idea of analyzing the influence of the architecture on model selection and generalization.

**S3.** Reviewer mCKs liked the idea of bridging the training loss with OOD generalization, and Reviewer keeA found our paper interesting as a whole.

**S4.** All reviewers agreed that our paper was clearly written and generally had a good presentation, figures in particular.

# Revisions

**R1.** **Characterization of the experimented distribution shifts (Reviewer U9Kt, keeA).** We followed the [official code](https://github.com/m-Just/OoD-Bench/tree/main) from [OODBench](https://openaccess.thecvf.com/content/CVPR2022/papers/Ye_OoD-Bench_Quantifying_and_Understanding_Two_Dimensions_of_Out-of-Distribution_Generalization_CVPR_2022_paper.pdf) to classify the 13 employed datasets across correlation-/diversitty-shift axes and added results to Appendix C.1.

**R2.** **Lack of novelty of our Research Question (RQ) (Reviewer 9YgD).** We added a description in the Related Work section regarding the works referenced by the reviewer, which better clarifies the novelty of our RQ. Those works perform model selection employing a validation signal while we make model selection from the training loss alone, as also acknowledged in **S1**.

**R3.** **Dependence between hyper-parameters ($\mathbf{h}$) and model parameters ($\mathbf{w}$) (Reviewer 9YgD).** We made more explicit (Section 2.1) our assumption that for a fixed architecture-HP-configuration pair, the learning process converges to a fixed parameter set $\mathbf{w}$, which is reasonable under the condition of a fixed optimizer since the predictions of a neural network do not vary significantly across repeated runs with the same HP configuration.

# Clarifications

**C1.** **Extending the evaluation to more complex real-world natural images (Reviewer U9Kt).** We clarified that several of our benchmarks represent real-world natural images, e.g., the ISIC2018 dataset was part of an imaging challenge in MICCAI 2018.

**C2.** **Novelty of architectural adjustments introduced (Reviewer 9YgD).** We clarified that the novelty does not come from the individual components but the architecture as a whole since architectures preserving variance were never proposed.

**C3.** **The small datasets employed do not qualify as an OOD scenario (Reviewer mCKs).** We clarified that learning from small/insufficient data has been cast as an OOD problem by [(Wad et al., 2022)](https://arxiv.org/pdf/2207.12258) as visible in Figure 2 from their work. Aggressively subsampling a large dataset incurs the learner to face an OOD scenario. We also provided additional evidence of this via **R1** since all the small datasets showed either diversity (more) or correlation (less) shifts.

**C4.** **Performance comparisons before and after architectural modifications (Reviewer keeA).** See Sections 2.2, 3.2, 3.3.

**C5.** **Key challenges to predict OOD performance in standard architectures (Reviewer keeA).** Pointed to Section 2.2.

**C6.** **Experiments on standard OOD datasets (Reviewer keeA, mCKs).** We employed popular OOD datasets such as CIFAR10-/100-C and domain-shifted versions of Tiny Imagenet (-C/-V2/-A/-R), following the Imagenet ones (due to computational constraints).

**C7.** **Focus of this paper is to be moved into the HP optimization literature (Reviewer keeA).** Our paper dedicated a Related Work paragraph to HP optimization, where we discussed its position and tested extensive HP optimization setups.

---

> ### Author Response · Authors · 2024-12-04
> **Summary of Rebuttal period (1)**
>
> # Points of disagreement
>
> While we greatly value the reviewers’ perspectives, we respectfully disagree on the following points that we thoroughly discussed in the discussion threads:
>
> **D1.** **The method only prevents overfitting, and various methods could be used to avoid it (Reviewer mCKs).** In our opinion, the reviewer is missing two critical points which lead to the main contribution of our work:
>
> When performing model selection, standard practice in the OOD literature recommends: 1) training-domain validation set, 2) leave-one-domain-out cross-validation, or 3) test-domain validation set ([Gulrajani and Lopez-Paz, 2020](https://arxiv.org/pdf/2007.01434)). Therefore, the only realistic way to avoid overfitting is to split the available data (1) or leave one domain out if domain labels are available (2). Therefore, in theory, there exist methods that "prevent overfitting" and help predict generalization for model selection.
>
> However, validation data is not always reliable, especially for OOD generalization prediction, as shown in  [(Teney et al., 2024](https://arxiv.org/pdf/2209.00613)), overfitting may still appear because features that work for IID may be spurious for OOD generalization.
> Following this empirical evidence and motivated by the unreliability of IID validation data for OOD model selection, we asked ourselves if it is possible to more reliably predict OOD generalization without the need for any IID validation data. Therefore, our contribution, acknowledged in **S1**, and consequent adjustments to the architecture prevent overfitting and enable reliable predictability (Table 1).
>
> **D2.** **Focusing only on stability makes the method ineffective and causes a performance drop (Reviewer mCKs, 9YgD).**
>
> While we generally agree that our method in the current form causes a trade-off between predictability and generalization, **we noticed that when the number of classes is small < 15, we reach equivalent accuracy to validation-based model selection, while when the number of classes rises, we observe a performance gap.**
>
> Our study, as discussed in Section 3.3, does not tell the whole story since we are the first ones to investigate such a solution, and our first contribution is to show that architecture design plays an important role in the prediction of OOD alignment, which is understudied for OOD generalization prediction.

---

### Meta-Review · Area_Chair_rDRX · 2024-12-23

**Metareview:**

This paper explores the potential for the training loss to predict out-of-distribution (OOD) generalization performance. It does this through a specially designed neural network architecture that aims to control prediction variance across distributions, motivated by a correlation-based analysis of training loss and test performance. The proposed architecture is then tested on a range of OOD datasets where it is shown to indeed have high correlation between training loss and OOD performance.

Strengths: The paper proposes an interesting and novel approach to an important topic, and is well presented.

Weaknesses: The proposed architecture suffers a large (> 5%) performance drop in many cases, and the focus on small data settings may conflate the effects of "overfitting" due to model estimation error (lack of data) vs domain shift, especially in the case study. There is also limited evaluation on real-world OOD datasets.

Overall, while the paper presents an interesting research direction, the practical impact of the work is somewhat unclear due to the large performance drops of the proposed architecture. Also, as OOD generalization generally requires assumptions, it would be useful to understand the conditions under which the proposed architecture works (in the sense of having correlated training and OOD loss) and does not; analysis of its performance on a broader range of real-world datasets will help address this. The AC believes this is an interesting direction of work but in its current form is not quite ready for publication, and hopes the comments will help strengthen the paper for a future submission.

**Additional Comments On Reviewer Discussion:**

One key issue raised was that the paper focuses on achieving a high correlation between the training and OOD losses but does not also take into account the model's performance, as there are large performance drops observed in many cases. The authors acknowledged the tradeoff but reviewers (mCKs and 9YgD) felt this was still a key concern after the discussion.

Another key issue discussed was regarding the small data regime used in the case study and how the results suggest that overfitting (in the sense of model estimation error due to lack of data) is the cause of the lack of correlation. There was quite a bit of discussion about this but the reviewers (mCKs and U9Kt) remained unconvinced by the authors' responses.

Finally, there were requests for evaluations on more real-world OOD datasets (U9Kt, mCKs, keeA) but the authors argued that they have already chosen a representative set of datasets in their experiments.

There were also several questions about novelty of the work, the problem setting and the types of shifts that the method works with that were addressed by the authors' rebuttal.

Overall, the large performance drops diminish the practical impact of the work (major) and the choice of small data for the case study confuses the presentation of the contribution. Having more evaluations on real-world OOD datasets will help to better characterize the proposed method/architecture as highlighted above.

---

### Decision · Program_Chairs · 2025-01-22

Reject